# Deep End-to-end Causal Inference

**Tomas Geffner**[† 1 *]   **Javier Antoran**[† 2 *]   **Adam Foster**[3 *]   **Wenbo Gong**[3]   **Chao Ma**[3]
**Emre Kiciman**[3]   **Amit Sharma**[3]   **Angus Lamb**[† 4]   **Martin Kukla**[3]
**Nick Pawlowski**[3]   **Agrin Hilmkil**[3]   **Joel Jennings**[3]
**Meyer Scetbon**[3]   **Miltiadis Allamanis**[3]   **Cheng Zhang**[3]
**[1] University of Massachusetts Amherst       [2] University of Cambridge**
**[3] Microsoft Research       [4] G-Research**
WenboGong@microsoft.com

Reviewed on OpenReview: `https://openreview.net/forum?id=e6sqttxEGX`

## Abstract

Causal inference is essential for data-driven decision-making across domains such as business engagement, medical treatment, and policy making. However, in practice, causal inference suffers from many limitations including unknown causal graphs, missing data problems, and mixed data types. To tackle those challenges, we develop Deep End-to-end Causal Inference (DECI) framework, a flow-based non-linear additive noise model combined with variational inference, which can perform both Bayesian causal discovery and inference. Theoretically, we show that DECI unifies many existing structural equation model (SEM) based causal inference techniques and can recover the ground truth mechanism under standard assumptions. Motivated by the challenges in the real world, we further extend DECI to heterogeneous, mixed-type data with missing values, allowing for both continuous and discrete treatment decisions. Empirically, we conduct extensive experiments (over a thousand) to show the competitive performance of DECI when compared to relevant baselines for both causal discovery and inference with both synthetic and causal machine learning benchmarks across data types and levels of missingness.

## 1 Introduction

Causal-aware decision making is pivotal in many fields such as economics (Zhang & Chan, 2006; Battocchi et al., 2021) and healthcare (Bica et al., 2019; Huang, 2021; Tu et al., 2019). For example, in healthcare, caregivers may wish to understand the effectiveness of different treatments given only historical data. They aspire to estimate treatment effects from the observational data, which we refer to as *causal inference task*. Unfortunately, there exist several challenges that prevent standard causal inference techniques to be applied in the above real-world scenario.

First, existing causal inference methods for estimating causal quantities from data commonly assume complete *a priori* knowledge of the causal relations (i.e. *causal graph*), which is rarely known, especially with many variables. Secondly, there may exist missing values and mixed data types in the history dataset, which poses another layer of difficulty to apply the causal inference methods.

To tackle the above challenges without latent confounders, we propose a *Deep End-to-end Causal Inference* (DECI) framework[1], a flow-based non-linear additive noise model combined with variational inference, where *end-to-end* represents the capability of performing both Bayesian causal discovery and inference under missing values and mixed-type data. Figure 1 illustrates the pipeline of DECI framework.

To enable this, we specifically contribute the following:

---

*Equal contribution. † Contributed during internship or residency in Microsoft Research.
[1]Code for reproducing experiments is available at `https://github.com/microsoft/causica/tree/v0.0.0`.
  Latest version of DECI is `https://github.com/microsoft/causica`.

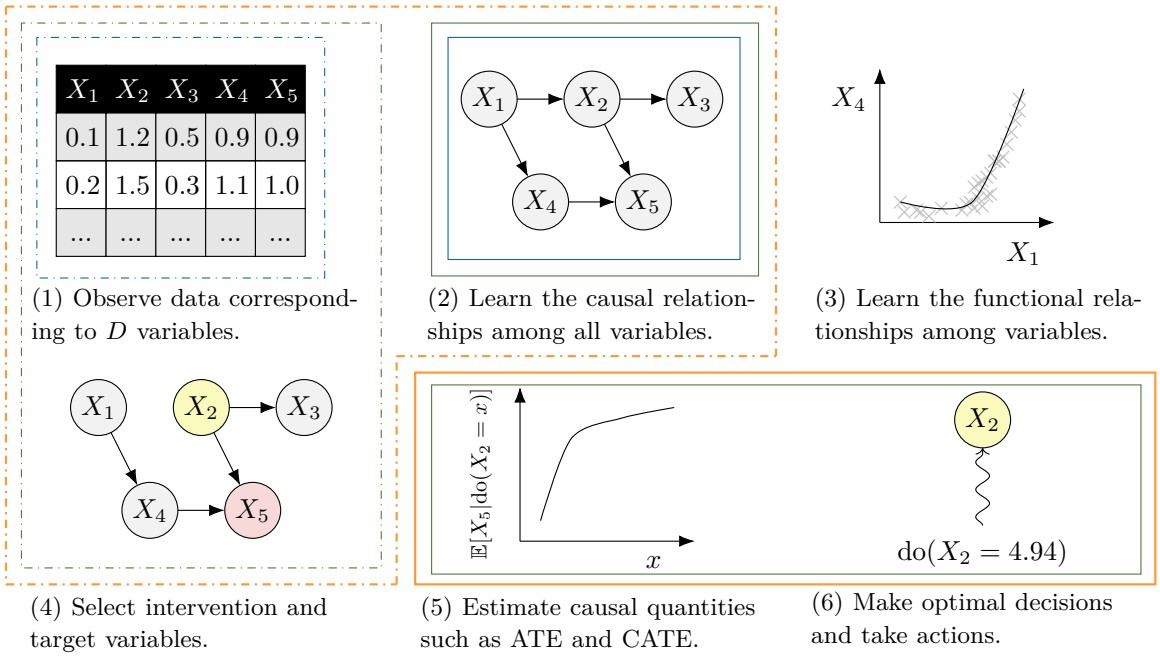

(1) Observe data corresponding to $D$ variables.

(2) Learn the causal relationships among all variables.

(3) Learn the functional relationships among variables.

(4) Select intervention and target variables.

(5) Estimate causal quantities such as ATE and CATE.

(6) Make optimal decisions and take actions.

Figure 1: An overview of the Deep End-to-end Causal Inference pipeline compared to traditional causal discovery and causal inference. The dashed line boxes show the inputs and the solid line boxes show the outputs. In causal discovery, a user provides observational data (1) as input. The output is the causal relationship (2) which are either DAGs or partial DAGs. In causal inference, the user needs to provide both the data (1) and the causal graph (2) as input and provide a causal question by specifying treatment and effect (4). Then, a model is learned and outputs the causal quantities (5), which helps decision making (6). In this work, the proposed DECI can perform both causal discovery and inference, which only requires the user to provide the observational data (1) and specify any causal questions (4), and output both the discovered causal relationship (2) and the causal quantities (5) that help in decision making (6).

1. *DECI formulation for discovery and inference.* We propose an end-to-end framework that combines non-linear additive noise SEM with variational inference to infer the graph posterior and learn functional parameters. Further, we extend the noise distribution beyond standard Gaussian and propose a flexible noise based on spline transformation (Durkan et al., 2019). With the learned SEM, we derive how to use it for simulation-based treatment effect estimation (i.e. (conditional) average treatment effect ((C)ATE)).

2. *Theoretical analysis of DECI.* To show the theoretical advantages of DECI, we first prove that under correct model specification and certain assumptions, DECI asymptotically recovers the true data-generating mechanism and correctly estimates the treatment effect. Additionally, we show DECI is a flexible model that unifies many existing SEM based causal methods, such as *Notears* (Zheng et al., 2018a; 2020), *Grandag* (Lachapelle et al., 2019), *MissDAG* (Gao et al., 2022), and others (Ng et al., 2019; 2020)

3. *Extension of DECI to real-world scenario.* Due to the unifying framework, we can equip DECI to perform missing value imputation similar to *MissDAG*. In addition, we also show how to handle mixed-type data (continuous and categorical).

4. *Extensive empirical evaluations.* To verify the effectiveness of DECI, we conduct extensive experiments (including synthetic and real-world benchmarks), along with a range of existing discovery and inference algorithms. DECI consistently outperforms the baselines in the discovery tasks and performs very competitively in causal discovery + inference tasks.

## 2 Related Work and Preliminaries

**Related Work.** Our work is related to both causal discovery and causal inference research. Approaches for causal discovery from observational data can be classified into three groups: constraint-based, score-based, and functional causal models (Glymour et al., 2019). Recently, Zheng et al. (2018a) framed the directed acyclic graph (DAG) structure learning problem as a continuous optimization task. Extensions (Lachapelle et al., 2019; Zheng et al., 2020; Gao et al., 2022) employ nonlinear function approximators, like neural networks, to model the relationships among connected variables. Our work combines this class of approaches with standard causal assumptions (Ng et al., 2019), to obtain our main theorem about causal graph learning. We extend functional methods to handle mixed data types and missing values. In particular, *MissDAG* proposed to use the expectation maximisation framework to perform missing value imputation and causal discovery at the same time. It represents a special case of DECI with Gaussian noise and point estimate graph. Therefore, we also propose the similar technique for imputing missing values. Outside of functional causal discovery, functional relationships between variables (see Figure 1(3)) are typically not learned by discovery algorithms (Spirtes & Glymour, 1991). Thus, distinct models, with potentially incompatible assumptions or inputs, must be relied upon for causal inference. However, when a DAG cannot be fully identified given the available data, constraint and score-based methods often return partially directed acyclic graphs (PDAGs) or completed partially directed acyclic graphs (CPDAGs) (Chickering, 2002; Andersson et al., 1997). Another line of works consists in estimating a posterior distribution over the DAGs from observed data. The central challenge in this Bayesian setting (Friedman & Koller, 2003; Heckerman et al., 2006; Tong & Koller, 2001) lies in the support of the posterior which is a union of exponentially growing (discrete) DAGs. Prior works have used Markov Chain Monte Carlo (MCMC) to directly sample DAGs or bootstrap traditional discovery methods (Chickering, 2002; Murphy, 2001; Tong & Koller, 2001), but these methods are typically limited to linear models which admit closed-form marginalization over continuous parameters. Recent advances have begun to utilize gradient information for more efficient inference (Lorch et al., 2021; Gong et al., 2021). In this work, we leverage these recent advances in DECI to estimate a distribution over DAGs.

Causal inference methods assume that either the graph structure is provided (Pearl, 2009b) or relevant structural assumptions are provided without the graph (Imbens & Rubin, 2015). Causal inference can be decomposed into two steps: identification and estimation. Identification focuses on converting the causal estimand (e.g. $P(Y|\text{do}(X = x), W)$) into an estimand that can be estimated using the observed data distribution (e.g. $P(Y|X, W)$). Common examples of identification methods include the back-door and front-door criteria (Pearl, 2009b), and instrumental variables (Angrist et al., 1996). Causal estimation computes the identified estimand using statistical methods, such as simple conditioning, inverse propensity weighting (Li et al., 2018), or matching (Rosenbaum & Rubin, 1983; Stuart, 2010). Machine learning-based estimators for CATE have also been proposed (Chernozhukov et al., 2018; Wager & Athey, 2018). Recent efforts to weaken structural assumption requirements (Guo & Perkovic, 2021; Jung et al., 2021) allow to perform causal inference for PDAGs and CPDAGs. However, both require prerequisite graphs as input (i.e. partially ancestral graph or maximally oriented PDAG), whereas DECI can infer a distribution over DAGs from observational data. In addition, DECI, once learned from data with satisfied assumptions, can provide identification on all causal quantities, while Guo & Perkovic (2021); Jung et al. (2021) either assumes identifiability at the beginning or provides a set identification on total causal effect. Jung et al. (2021) also leverage double machine learning (Chernozhukov et al., 2018), a linear model, as the estimation for causal quantities, compared to the nonlinear neural network of DECI. Pawlowski et al. (2020) proposed a variational approach, DSCM, to infer the posterior noise distribution during counterfactual computation. However, they assume a causal graph as a priori and extend beyond the additive noise structure. Our method is orthogonal to it and can infer both causal graph and counterfactuals.

**Structural Equation Models (SEM).** Let $\mathbf{x} = (x_1, \ldots, x_D)$ be a collection of random variables. SEMs (Pearl, 2009b) model causal relationships between the individual variables $x_i$. Given a DAG $G$ on nodes $\{1, \ldots, D\}$, $\mathbf{x}$ can be described by $x_i = F_i\big(\mathbf{x}_{\text{pa}(i;G)}, z_i\big)$, where $z_i$ is an exogenous noise variable that is independent of all other variables in the model, $\text{pa}(i; G)$ is the set of parents of node $i$ in $G$, and $F_i$ specifies how variable $x_i$ depends on its parents and the noise $z_i$. In this paper, we focus on additive noise SEMs,

also referred to as additive noise models (ANM), i.e.

$$F_i\left(\mathbf{x}_{\mathrm{pa}(i;G)}, z_i\right) = f_i\left(\mathbf{x}_{\mathrm{pa}(i;G)}\right) + z_i \quad \text{or} \quad \mathbf{x} = f_G(\mathbf{x}) + \mathbf{z} \quad \text{in vector form.} \tag{1}$$

Here, we omit the dependency of function $f_i$ to its parameters $\theta$. To represent the graph $G$, we use binary adjacency matrix. In this paper, we interchangeably use $G$ to represent graph and adjacency matrix, where $G_{ij} = 1$ represents $i \to j$.

**Average Treatment Effects.** The ATE and CATE quantities allow us to estimate the impact of our actions (treatments) (Pearl, 2009b). Assume that $\mathbf{x}_T$ (with $T \subset \{1, \ldots, D\}$) are the treatment variables; the interventional distribution is denoted $p(\mathbf{x} \mid \mathrm{do}(\mathbf{x}_T = \mathbf{a}))$. The ATE and CATE on targets $\mathbf{x}_Y$ for treatment $\mathbf{x}_T = \mathbf{a}$ given a reference $\mathbf{x}_T = \mathbf{b}$, and conditional on $\mathbf{x}_C = \mathbf{c}$ for CATE, are given by

$$\mathrm{ATE}(\mathbf{a}, \mathbf{b}) = \mathbb{E}_{p(\mathbf{x}_Y|\mathrm{do}(\mathbf{x}_T=\mathbf{a}))}[\mathbf{x}_Y] - \mathbb{E}_{p(\mathbf{x}_Y|\mathrm{do}(\mathbf{x}_T=\mathbf{b}))}[\mathbf{x}_Y], \quad \text{and} \tag{2}$$

$$\mathrm{CATE}(\mathbf{a}, \mathbf{b}|\mathbf{c}) = \mathbb{E}_{p(\mathbf{x}_Y|\mathrm{do}(\mathbf{x}_T=\mathbf{a}),\mathbf{x}_C=\mathbf{c})}[\mathbf{x}_Y] - \mathbb{E}_{p(\mathbf{x}_Y|\mathrm{do}(\mathbf{x}_T=\mathbf{b}),\mathbf{x}_C=\mathbf{c})}[\mathbf{x}_Y]. \tag{3}$$

We consider the common scenario where the *conditioning variables are not caused by the treatment*. Sepcifically, we focus on the scenario where the treatment $\mathbf{x}_T$ is not the cause of the conditioning set $\mathbf{x}_C$, i.e. there is no directed path from $T$ to $C$ in $G$.

## 3 DECI: Deep End-to-end Causal Inference

We introduce DECI, an end-to-end deep learning-based causal inference framework. DECI learns a distribution over causal graphs from observational data and (subsequently) estimates causal quantities. Section 3.1, describes our autoregressive flow based ANM SEM. Section 3.2 lays out the conditions under which DECI will recover the true causal graph given enough observational data (Theorem 1). Section 3.3 shows how the generative model learnt by DECI can be used to simulate samples from intervened distributions, allowing for treatment effect estimation. Section 3.4 extends DECI's real-world applicability by adding support for non-Gaussian exogenous noise, mixed type data (continuous and discrete), and imputation for partially observed data.

### 3.1 DECI and Causal Discovery

DECI takes a Bayesian approach to causal discovery (Heckerman et al., 1999). We model the causal graph $G$ jointly with the observations $\mathbf{x}^1, \ldots, \mathbf{x}^N$ as

$$p_\theta(\mathbf{x}^1, \ldots, \mathbf{x}^N, G) = p(G) \prod_n p_\theta(\mathbf{x}^n|G). \tag{4}$$

We aim to fit $\theta$, the parameters of our non-linear ANM, using observational data. Once this model is fitted, the posterior $p_\theta(G|\mathbf{x}^1, \ldots, \mathbf{x}^N)$ characterizes our beliefs about the causal structure.

**Prior over Graphs.** The graph prior $p(G)$ should characterize the graph as a DAG. We implement the prior as

$$p(G) \propto \exp\left(-\lambda_s \|G\|_F^2 - \rho\, h(G)^2 - \alpha\, h(G)\right), \tag{5}$$

where we weight the DAG penalty by $\alpha$ and $\rho$, $\|\cdot\|_F$ indicates the Frobenius norm and $h(G) = \mathrm{tr}(e^{G \odot G}) - D$ with $\odot$ is the Hadamard product. The squared term is from the augmented Lagrangian (Nemirovsky, 1999). The advantage is that it closely approximates the solution of the constraint problem with the solution of unconstrained problem without increasing $\rho$ to infinity. $\rho$, $\alpha$ are gradually increased during training following an augmented Lagrangian scheme, ensuring only DAGs remain at convergence. We introduce prior knowledge about graph sparseness by penalising the norm $\lambda_s \|G\|_F$, with $\lambda_s$ a scalar. We can also incorporate domain knowledge to specify a more informative prior, see appendix E.6 for full details.

**Likelihood of Structural Equation Model.** Following Khemakhem et al. (2021), we factorize the observational likelihood $p_\theta(\mathbf{x}^n|G)$ in an autoregressive manner. Rearranging the ANM assumption eq. (1), we have $\mathbf{z} = g_G(\mathbf{x}; \theta) = \mathbf{x} - f_G(\mathbf{x}; \theta)$. The components of $\mathbf{z}$ are independent. If we have a distribution $p_{z_i}$ for

component $z_i$ where $z_i$ is the exogenous noise for variable $x_i$, then we can write the observational likelihood as

$$p_\theta(\mathbf{x}^n|G) = p_\mathbf{z}\left(g_G(\mathbf{x}^n;\theta)\right) = \prod_{i=1}^{D} p_{z_i}\left(g_{G,i}(\mathbf{x}^n;\theta)\right), \tag{6}$$

where the first equality is from the change of variable formula since $g_G$ is the invertible transformation between $\mathbf{x}, \mathbf{z}$ and we omitted the Jacobian-determinant term because it is always equal to one for DAGs $G$ (lemma 3 in appendix C). $g_{G,i}$ represents the $i^{\text{th}}$ elements of its output. The choice of $f_G : \mathbb{R}^d \to \mathbb{R}^d$ must satisfy the adjacency relations specified by the graph $G$. If there is no edge $j \to i$ in $G$, then the function $f_i(\mathbf{x})$—the $i$-th component of the output of $f_G(\mathbf{x})$— must satisfy $\partial f_i(\mathbf{x})/\partial x_j = 0$. We propose a flexible parameterization that satisfies this by setting

$$f_i(\mathbf{x}) = \zeta_i\left(\sum_{j=1}^{d} G_{j,i}\ \ell_j(x_j)\right), \tag{7}$$

where $G_{j,i} \in \{0,1\}$ indicates the presence of the edge $j \to i$, and $\ell_i$ and $\zeta_i$ $(i = 1, \ldots, d)$ are MLPs. A naïve implementation would require training $2D$ MLPs. Instead, we construct these MLPs so that their weights are shared across nodes as $\zeta_i(\cdot) = \zeta(\mathbf{u}_i, \cdot)$ and $\ell_i(\cdot) = \ell(\mathbf{u}_i, \cdot)$, with $\mathbf{u}_i \in \mathbb{R}^D$ a trainable embedding that identifies the source and target nodes respectively.

**Exogenous Noise Model $p_\mathbf{z}$.** We consider two possible models for the distribution of $\mathbf{z}$: 1) A simple Gaussian $p_{z_i}(\cdot) = \mathcal{N}\left(\cdot|0, \sigma_i^2\right)$, where per-variable variances $\sigma_i^2$ are learnt. 2) A flow (Rezende & Mohamed, 2015)

$$p_{z_i}(z_i) = \mathcal{N}\left(\kappa_i^{-1}(z_i)|0, 1\right)\left|\frac{\partial \kappa_i^{-1}(z_i)}{\partial z_i}\right|. \tag{8}$$

We choose the learnable bijections $\kappa_i$ to be a rational quadratic spline (Durkan et al., 2019), parametrised independently across dimensions. We do not couple across dimensions since our SEM requires independent noise variables. Spline flows are significantly more flexible than the Gaussian distributions employed in previous work (Lachapelle et al., 2019; Ng et al., 2019; 2020; Zheng et al., 2018a; 2020). Here, we use Gaussian distribution as the base, but one can easily replace it with other forms of distributions, as long as there exists differentiable density functions.

**Optimization and Inference Details.** The described model presents two challenges. First, the true posterior over $G$ is intractable. Second, maximum-likelihood cannot be used to fit the model parameters, due to the presence of the latent variable $G$. We simultaneously overcome both of these challenges using variational inference (Blei et al., 2017; Jordan et al., 1999; Zhang et al., 2018). We define a variational distribution $q_\phi(G)$ to approximate the intractable posterior $p_\theta(G|\mathbf{x}^1, \ldots, \mathbf{x}^N)$, and use it to build the ELBO, given by

$$\text{ELBO}(\theta, \phi) = \mathbb{E}_{q_\phi(G)}\left[\log p(G)\prod_n p_\theta(\mathbf{x}^n|G)\right] + H(q_\phi) \leq \log p_\theta(\mathbf{x}^1, \ldots, \mathbf{x}^N), \tag{9}$$

where $H(q_\phi)$ represents the entropy of the distribution $q_\phi$ and $p_\theta(\mathbf{x}^n|G)$ takes the form of eq. (6) (derivation in Appendix B.3). We model $q_\phi(G)$ as the product of independent Bernoulli distributions, one for each potential directed edge in $G$. We use the ENCO parameterization (Lippe et al., 2021) to represent the probability of each potential directed edge as $q_\phi(G) = \prod_{ij} \text{Ber}(G_{ij} \mid \sigma(\gamma_{ij})\sigma(\vartheta_{ij}))$ where $\sigma(\cdot)$ is the sigmoid function, $\gamma \in \mathbb{R}^{D \times D}$ represents the existence of an edge, and $\vartheta \in \mathbb{R}^{D \times D}$ with $\vartheta_{ij} = -\vartheta_{ji}$ describes the edge orientation. $\sigma(\cdot)$ indicates the sigmoid function. Therefore, the two orientation probabilities will always sum to one. A high probability direction indicates a low probability of the other direction. This can help to prevent the length-1 cycles. In addition, the existence probability is easier to learn compared to the orientation since it does not need to consider dagness. Thus, the separation of existence and orientation probability allows for better control of gradient for easier optimization. The SEM parameters $\theta$ and variational parameters $\phi$ are trained by maximizing the ELBO. The Gumbel-softmax trick (Jang et al., 2016; Maddison et al., 2016) is used to stochastically estimate the gradients with respect to $\phi$. Appendix B.1 details the full optimisation procedure.

**Unified View of Functional Causal Discovery.** We note that, like DECI, many functional causal discovery methods (Lachapelle et al., 2019; Ng et al., 2019; 2020; Zheng et al., 2018a; 2020) can be seen from a probabilistic perspective as fitting an autoregressive flow (with a hard acyclicity constraint) for different choices for the exogenous noise distribution $p_{\mathbf{z}}$ and transformation function $f$. We expand on the details of this perspective and formalise it in Appendix C. In a nutshell, we show that the training objective and model parametrization of many existing causal discovery methods can be viewed as a special case of DECI's parametrization and objective function. In particular,

***Notears* (Zheng et al., 2018a)** uses a standard isotropic Gaussian for $p_{\mathbf{z}}$ and a linear transformation for $f(\mathbf{x}, \theta)$.

***Notears-MLP* (Zheng et al., 2020)** uses a standard isotropic Gaussian for $p_{\mathbf{z}}$ and $D$ independent multi-layer perceptrons (MLP), one for each $f_i(\mathbf{x})$.

***Notears-Sob* (Zheng et al., 2020)** uses a standard isotropic Gaussian for $p_{\mathbf{z}}$ and a weighted linear combination of nonlinear basis functions for each $f_i(\mathbf{x})$.

***GAE* (Ng et al., 2019)** uses a standard isotropic Gaussian for $p_{\mathbf{z}}$ and a GNN for $f_i(\mathbf{x})$.

***Grandag* (Lachapelle et al., 2019)** uses a factorized Gaussian with mean zero and learnable scales for $p_{\mathbf{z}}$ and $D$ MLP, one for each $f_i(\mathbf{x})$.

***Golem* (Ng et al., 2020).** This is a linear method whose original formulation was already in a probabilistic perspective, using a linear transformation for $f_i(\mathbf{x})$.

Therefore, we can see DECI framework provided a unified view on the above methods. DECI also extends the above by allowing the $f$ and noise distribution to be MLP and potentially non-Gaussian, respectively, which makes it the most flexible member of this family.

## 3.2 Theoretical Considerations

One key statistical guarantee of DECI is to show that optimizing the variational objective eq. (9) leads to the recovery of the ground truth data generation mechanism under the infinite data limit.

We formalize this guarantee in Theorem 1, which states that under certain assumptions, eq. (9) is indeed a valid objective function. Specifically, Theorem 1 requires the following assumptions:

- *Non-trivial function*, the function $f$ is non-invertible, $3^{\text{rd}}$-order differentiable and is not constant w.r.t. any of its inputs. This assumption can be met by using MLPs.

- *Regularity of noise distribution*, the noise distribution has a differentiable probability density function with bounded likelihood.

- *Correct specification*, there exists a parameter $\theta^*$, such that $p_{\theta^*}(\boldsymbol{x}; G^0)$ is the data-generating distribution, where $G^0$ is the ground truth causal graph. Note that although $\theta^*$ may not be equal to the ground truth parameter, we do not care about the identifiability in parameter spaace, but in functional space. Namely, the equality of functions.

- *No latent confounder*, this is also called causal sufficiency where there are no un-observed confounders in the data.

The above assumptions in fact imply some of the common causal assumptions, e.g. Markov property, causal minimality, and exclude some abnormal cases (e.g. linear Gaussian SEMs) for model structural identifiability. Refer to Appendix A for formal statements and proof.

**Theorem 1** (DECI recovers the true data generating process). *Assume DECI's underlying SEM belongs to continuous additive noise structural equation models with (1) non-invertible, $3^{rd}$-order differentiable function between variables that are not constant w.r.t. any of their inputs (i.e. non-trivial); (2) differentiable noise density and bounded log-likelihood, then, under a correctly specified model, infinite data limit and no latent confounders in data generating mechanism, the solution $(\theta', q'_\phi(G))$ from maximizing the ELBO (eq. (9)) satisfies $q'_\phi(G) = \delta(G = G')$ where $G'$ is a unique graph. In particular, $G' = G^0$ and $p_{\theta'}(\mathbf{x}; G') = p(\mathbf{x}; G^0)$.*

$\delta$ represents the Dirac delta function. Note that we do not claim the identifiability of $\theta$ in the parameter space, but rather on the functional space. To demonstrate the trend of graph posterior converging to a single graph with increasing dataset size, we plot its entropy with different dataset sizes on Figure 6 in appendix A.5.

Note that Theorem 1 only proves that DECI can recover the true graph and data generating distributions when ELBO is maximized under infinite data regime. We do not prove the tendency on whether the optimization can maximize the ELBO, which depends on properties of the chosen optimization and is out of the scope of this work. One potential future direction to strengthen Theorem 1 is to prove the convergence rate w.r.t the number of data points, similar to the variational Bernstein Von-Mises theorem (Wang & Blei, 2018). Another point worth noting is that Theorem 1 does not require the uniqueness of the model parameters $\theta$, as we only prove the recovery in the distributional sense. The claim is still valid even if we have a set of parameters, as long as they induce the same distribution as the ground truth distribution under the true graph.

**Model Misspecification**  One of the key assumption for theorem 1 is the model specification. When the model is misspecified, theorem 1 no longer holds. This means that there is no guarantee on the recovery of the true generation mechanism and graph. There are some existing theoretical works on variational inference under the misspecified model (Wang & Blei, 2019; Knoblauch, 2019). Wang & Blei (2019) claims that even with model misspecification, the inferred posterior will collapse to a point that minimizes the KL divergence between the resulting and ground-truth generation mechanism. However, their methodology cannot be directly applied to our setting, since they assume that the random variables are continuous rather than discrete like graphs. This opens a new research opportunity to prove a similar guarantee for graphs as well. Despite that, we can still leverage the universal approximation capacity of MLPs (Hornik et al., 1989), meaning that they can approximate continuous functions arbitrarily well. This flexibility gives us a higher chance that this assumption indeed holds.

### 3.3  Estimating Causal Quantities

We now show how the generative model learned by DECI can be used to evaluate expectations under interventional distributions, and thus estimate ATE and CATE. As explained above, DECI returns $q_\phi(G)$, an approximation of the posterior over graphs given observational data. Then, interventional distributions and treatment effects can be obtained by marginalizing over graphs as

$$\mathbb{E}_{q_\phi(G)}\left[p\left(\mathbf{x}_Y|\mathrm{do}(\mathbf{x}_T{=}\mathbf{a}), G\right)\right], \quad \mathbb{E}_{q_\phi(G)}[\mathrm{ATE}(\mathbf{a}, \mathbf{b}|G)]$$
$$\text{and} \quad \mathbb{E}_{q_\phi(G)}[\mathrm{CATE}(\mathbf{a}, \mathbf{b}|\mathbf{c}, G)]. \tag{10}$$

This can be seen as a probabilistic relaxation of traditional causal quantity estimators. When one has observed enough data to be certain about the causal graph, i.e., $q_\phi(G) = \delta(G - G_i)$, our procedure matches traditional causal inference. We go on to discuss how DECI estimates (C)ATE.

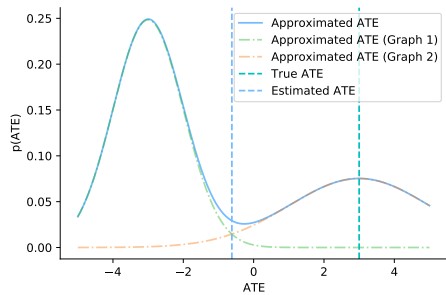

Figure 2: Marginalizing ATE estimates from graph and exogenous noise samples.

**Estimating ATE.**  After training, we can use the model learnt by DECI to simulate new samples $\mathbf{x}$ from $p_\theta(\mathbf{x}|G)$. We sample a graph $G \sim q_\phi(G)$, and a set of exogenous noise variables $\mathbf{z} \sim p_\mathbf{z}$. We then input this noise into the learnt DECI structural equation model to simulate $\mathbf{x}$, by applying eq. (1) and eq. (7) on $\mathbf{z}$ in the topological order defined by $G$. However, ATE estimation requires samples from the interventional distribution $p\left(\mathbf{x}_{\backslash T}|\mathrm{do}(\mathbf{x}_T = \mathbf{b}), G\right)$. These can be obtained by noting that

$$p\left(\mathbf{x}_{\backslash T}|\mathrm{do}(\mathbf{x}_T = \mathbf{b}), G\right) = p\left(\mathbf{x}_{\backslash T}|\mathbf{x}_T = \mathbf{b}, G_{\mathrm{do}(\mathbf{x}_T)}\right),$$

where $G_{\mathrm{do}(\mathbf{x}_T)}$ is the "mutilated" graph obtained by removing incoming edges to $\mathbf{x}_T$. Thus, samples from this distribution can be obtained by following the sampling procedure explained above, but fixing the values

$\mathbf{x}_T = \mathbf{b}$ and using $G_{\mathrm{do}(\mathbf{x}_T)}$ instead of $G$. Finally, we use these samples to obtain a Monte Carlo estimate of the expectations required for ATE computation eq. (2). When the posterior $q_\phi(G)$ has not collapsed to one graph, these samples are from a mixture distribution, translating the uncertainty in the discovered graph into uncertainty in ATE estimates. This is illustrated in Figure 2.

**Estimating CATE.** Under the assumption that $\mathbf{x}_T$ is not a cause of $\mathbf{x}_C$, we can estimate CATE by sampling from the interventional distribution $p\left(\mathbf{x}_{\backslash T}|\mathrm{do}(\mathbf{x}_T), G\right)$ and then estimating the conditional distribution of $\mathbf{x}_Y$ given $\mathbf{x}_C$. To make this precise, we let $Y = X \setminus (T \cup C)$ denote all variables that we do not intervene or condition on. Conditional densities

$$p_\theta(\mathbf{x}_Y \mid \mathrm{do}(\mathbf{x}_T{=}\mathbf{b}), \mathbf{x}_C{=}\mathbf{c}, G) = \frac{p_\theta(\mathbf{x}_Y, \mathbf{x}_C{=}\mathbf{c}|\mathbf{x}_T{=}\mathbf{b}, G_{\mathrm{do}(\mathbf{x}_T)})}{p_\theta(\mathbf{x}_C{=}\mathbf{c}|\mathbf{x}_T{=}\mathbf{b}, G_{\mathrm{do}(\mathbf{x}_T)})} \tag{11}$$

are not directly tractable in the DECI model due to the intractability of the marginal $p_\theta(\mathbf{x}_C{=}\mathbf{c}|\mathbf{x}_T{=}\mathbf{b}, G_{\mathrm{do}(\mathbf{x}_T)})$. However, we can always sample from the joint interventional distribution $p_\theta(\mathbf{x}_Y, \mathbf{x}_C|\mathbf{x}_T{=}\mathbf{b}, G_{\mathrm{do}(\mathbf{x}_T)})$. We use samples from this joint to train a surrogate regression model $h_G{}^2$ to the relationship between $\mathbf{x}_C$ and $\mathbf{x}_Y$. Specifically, we minimize the square loss

$$\mathbb{E}_{p_\theta(\mathbf{x}_Y, \mathbf{x}_C|\mathbf{x}_T{=}\mathbf{b}, G_{\mathrm{do}(\mathbf{x}_T)})}\left[\|\mathbf{x}_Y - h_G(\mathbf{x}_C)\|^2\right].$$

making $h_G$ approximate the conditional mean of $\mathbf{x}_Y$. We choose $h_G$ to be a basis-function linear model with random Fourier basis functions (Yu et al., 2016). As illustrated, in Figure 3, we train two separate surrogate models, one for our intervention $\mathbf{x}_T = \mathbf{a}$ and one for the reference $\mathbf{x}_T = \mathbf{b}$. We estimate CATE as the difference between their outputs evaluated at $\mathbf{x}_C = \mathbf{c}$. This process is repeated for multiple posterior graph samples $G \sim q_\phi$, allowing us to marginalise the posterior graphs

$$\mathbb{E}_{q_\phi(G)}\left[g_{G_{\mathrm{do}(\mathbf{x}_T=\mathbf{a})}}(\mathbf{x}_C = \mathbf{c}) - g_{G_{\mathrm{do}(\mathbf{x}_T=\mathbf{b})}}(\mathbf{x}_C = \mathbf{c}))\right]. \tag{12}$$

**Theoretical considerations.** We previously showed that variational inference asymptotically recovers the correct SEM parameters $\theta^*$ in Theorem 1. We further show that a correctly specified DECI model satisfies causal effect identifiability and positivity, meaning that treatment effects will be correctly estimated in the large data limit (refer to Appendix A.4).

**General ECI Framework.** The probabilistic treatment of the DAG, and the re-use of functional causal discovery generative models for simulation-based causal inference are principles that can be applied beyond DECI. Constraint-based (Spirtes & Glymour, 1991) and score-based (Chickering, 2002) discovery methods often output a set of DAGs compatible with the data, i.e., a PDAG or CPDAG. It is natural to interpret these equivalence classes as uniform distributions over members of sets of graphs. We can then use eq. (10) to estimate causal quantities by marginalizing over these distributions. The quantities inside the expectations over graphs can be estimated using any existing causal inference method, such as linear regression (Sharma et al., 2021), Double ML (Chernozhukov et al., 2018), etc. Our experiments explore combinations of discovery methods that return graph equivalence classes with standard causal inference methods. We take expectations over causal graphs since these return the quantity that minimises the posterior expected squared error in our (C)ATE estimates while noting that the best statistic will be application dependent.

### 3.4 DECI for Real-world Heterogeneous Data

We extend DECI to handle mixed-type (continuous and discrete) data, and data with missing values, which often arise in real-world applications.

**Handling Mixed-type Data.** For discrete-valued variables, we remove the additive noise structure and directly parameterise parent-conditional class probabilities

$$p_\theta^{\mathrm{discrete}}\left(x_i|\mathbf{x}_{\mathrm{pa}(i;G)}; G\right) = P_i\left(\mathbf{x}_{\mathrm{pa}(i;G)}; \theta\right)(x_i), \tag{13}$$

---

[2]Subscript $G$ allows differentiating surrogate models fit on samples from different graphs drawn from $q_\phi$.

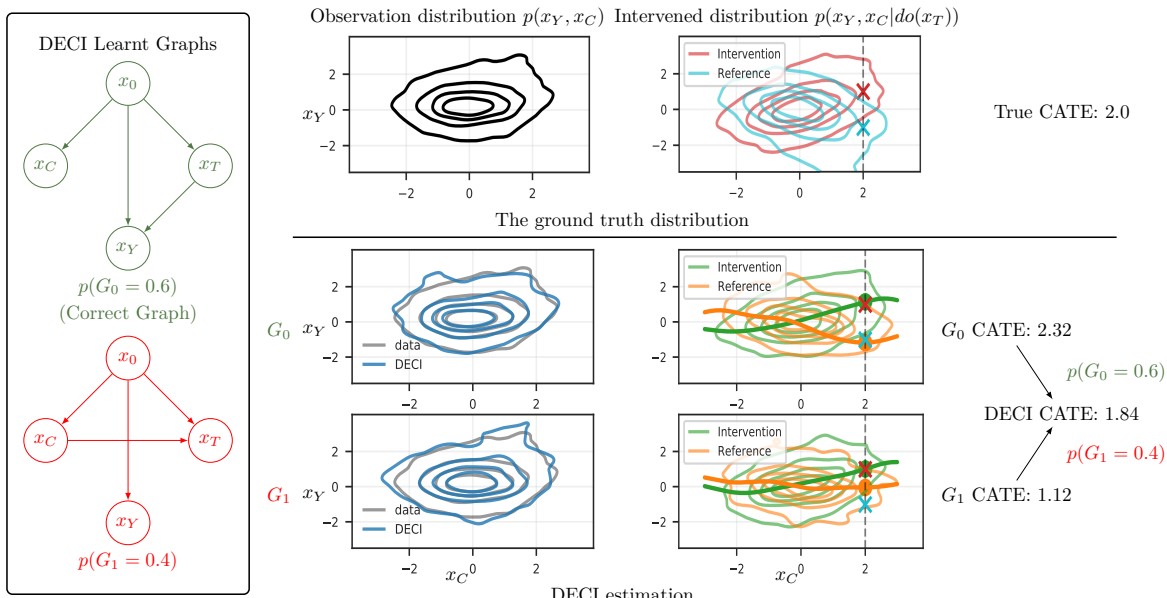

Figure 3: DECI CATE estimation on the CSuite Symprod Simpson dataset. Left: The DECI graph posterior has two modes with $p(G) = 0.6$ for the correct graph and $p(G) = 0.4$ for an alternative possibility with some incorrect edges. Middle: we display the joint distribution of conditioning and effect variables in the observational setting and under interventions on $\mathbf{x}_T$. DECI captures the observational density well. Right: interventional distributions with their conditional means $\mathbf{x}_C = \mathbf{c}$ marked with crosses. DECI predicts conditional expectations by fitting functions from $\mathbf{x}_C$ to $\mathbf{x}_Y$ and evaluating them at $\mathbf{c}$. DECI outputs CATE by marginalizing the result over possible graphs.

where $P_i\left(\mathbf{x}_{\mathrm{pa}(i;G)}; \theta\right)$ is a normalised probability mass vector over the number of classes of $x_i$, obtained by applying the softmax operator to $f_i(\mathbf{x}_{\mathrm{pa}(i;G)})$. This means that for discrete variables, the output of $f_i$ is a vector of length equal to the number of classes for variable $i$. This approach gives a valid likelihood for $p_\theta(\mathbf{x}^n|G)$ which we use to train DECI. However, since the full generative model is no longer an ANM, we cannot guarantee that Theorem 1 applies in this setting.

**Handling Missing Data.** We propose an extension of DECI to partially observed data.[3] We use $\mathbf{x}_o^n$ to denote the observed components of $\mathbf{x}^n$, $\mathbf{x}_u^n$ to denote the unobserved components, and their joint density in the observational environment is $p_\theta(\mathbf{x}_o^n, \mathbf{x}_u^n)$. We approximate the $p(G, \mathbf{x}_u^n|\mathbf{x}_o^n)$ with the variational distribution,

$$q_{\phi,\psi}\left(G, \mathbf{x}_u^1, \ldots, \mathbf{x}_u^N | \mathbf{x}_o^1, \ldots, \mathbf{x}_o^N\right) = q_\phi(G) \prod_n q_\psi(\mathbf{x}_u^n|\mathbf{x}_o^n),$$

which yields the following learning objective

$$\mathrm{ELBO}(\theta, \phi, \psi) = H(q_\phi) + \sum_n H(q_\psi(\mathbf{x}_u^n|\mathbf{x}_o^n)) + \mathbb{E}_{q_{\phi,\psi}}\left[\log p(G) \prod_n p_\theta(\mathbf{x}_o^n, \mathbf{x}_u^n|G)\right]. \tag{14}$$

We parameterize the Gaussian imputation distribution $q_{\psi_n}(\mathbf{x}_u^n|\mathbf{x}_o^n)$ using an amortization network (Kingma & Welling, 2013), whose input is $\mathbf{x}_o^n$, and outputs the mean and variance of $q_\psi(\mathbf{x}_u^n|\mathbf{x}_o^n)$.

# 4 Experiments

We evaluate DECI on both causal discovery and causal inference tasks. A full list of results and details of the experimental setup are in Appendices B and E.

---

[3]We assume that values are missing (completely) at random, the most common setting (Ma et al., 2018; Rubin, 1976; Stekhoven & Bühlmann, 2012; Strobl et al., 2018).

### 4.1 Causal Discovery Evaluation

**Datasets.**

We consider synthetic, pseudo-real, and real data. For the synthetic data, we follow Lachapelle et al. (2019) and Zheng et al. (2020) by sampling a DAG from two different random graph models, **Erdős-Rényi (ER)** and **scale-free (SF)**, and simulating each ANM $x_i = f_i(\mathbf{x}_{\mathrm{pa}(i;G)}) + z_i$, where $f_i$ is a nonlinear function (randomly sampled spline). We consider two noise distributions for $z_i$, a standard Gaussian and a more complex one obtained by transforming samples from a standard Gaussian with an MLP with random weights. We consider the number of nodes $d \in \{16, 64\}$ with the number of edges $e \in \{d, 4d\}$. The resulting datasets are identified as $\mathbf{ER}(d, e)$ and $\mathbf{SF}(d, e)$. All datasets have $n$=5000 training samples.

For the pseudo-real data we consider the **SynTReN** generator (Van den Bulcke et al., 2006), which creates synthetic transcriptional regulatory networks and produces simulated gene expression data that mimics experimental data. We use the datasets generated according to Lachapelle et al. (2019) ($d$=20), and take $n$=400 for training. Finally, for the real dataset, we use the observational protein measurements in human cells from Sachs et al. (2005), in which we sampled a training set with $n$=800 samples and $d$=11.

**Baselines.** We run DECI using two models for exogenous noise: a Gaussian with learnable variance (identified as DECI-G) and a spline flow (DECI-S). We compare against *PC* (Kalisch & Bühlman, 2007), (linear) *Notears* (Zheng et al., 2018a), the nonlinear variants *Notears-MLP* and *Notears-Sob* (Zheng et al., 2020), *Grandag* (Lachapelle et al., 2019), and *ICALiNGAM* (Shimizu et al., 2006). When a CPDAG is the output, e.g., from PC, we treat all possible DAGs under the CPDAG as having the same probability. All baselines are implemented with the `gcastle` package (Zhang et al., 2021).

**Causality Metrics.** We report F1 scores for adjacency, orientation (Glymour et al., 2019; Tu et al., 2019) and causal accuracy (Claassen & Heskes, 2012). For DECI, we report the expected values of these metrics estimated over the graph posterior.

Figure 4 shows the results for the data generated with non-Gaussian noise. We observe that DECI achieves the best results across all metrics. Additionally, using the flexible spline model for the exogenous noise (DECI-S) yields better results than the Gaussian model

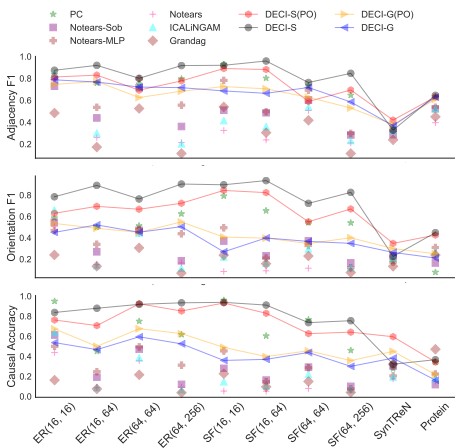

Figure 4: Causal discovery on benchmark datasets. The label *(PO)* corresponds to running DECI with 30% of the training data missing. For readability, the DECI results by are connected with with soft lines. The figure shows mean results across five different random seeds.

(DECI-G). This is expected, as the noise used to generate the data is non-Gaussian. For Gaussian noise (see Figure 8), both DECI-S and DECI-G perform similarly. Moreover, when data are partially observed *(PO)*, the strong performance of DECI remains, showing that DECI can handle missing data efficiently.

### 4.2 End-to-end Causal Inference

We evaluate the *end-to-end* pipeline, taking in observational data and returning (C)ATE estimates.

**Datasets.** We generate ground-truth treatment effects to compare against for the **ER** and **SF** synthetic graphs that were described in Section 4.1 by applying random interventions on these synthetic SEMs, ensuring at most 3 edges between the intervention and effect variables. For more detailed analysis, we hand-craft a suite of synthetic SEMs, which we name **CSuite**. CSuite datasets elucidate particular features of the model, such as identifiability of the causal graph, correct specification of the SEM, exogenous noise distributions, and size of the optimal adjustment set. We draw conditional samples from CSuite SEMs with Hamiltonian

Figure 5: Box plots showing end-to-end ATE and CATE estimation error on the semi-synthetic Twins and IHDP datasets with different method combinations. Method acronyms are as in Table 6. Box plots show the median and interquartile range over 5 seeds.

Monte Carlo (HMC) (Neal et al., 2011), allowing us to evaluate CATE. Finally, we include two semi-synthetic causal inference benchmark datasets for ATE evaluation: **Twins** (twin birth datasets in the US) (Almond et al., 2005) and **IHDP** (Infant Health and Development Program data) (Hill, 2011). See Appendix D for all experimental details.

**Baselines.**

To thoroughly evaluate end-to-end inference, we consider different ways of *combining* discovery and inference algorithms. For DECI, we can use a trained model to immediately estimate (C)ATE. We also consider using the learned DECI graph posterior in combination with existing methods for causal inference on a known graph: DoWhy-Linear (L) and DoWhy-Nonlinear (N) (Sharma et al., 2021) which implement linear adjustment and Double Machine Learning (DML) (Chernozhukov et al., 2018) methods for backdoor adjustment respectively. We also pair other *discovery* methods with DECI and DoWhy treatment effect estimation, namely the PC algorithm as a baseline and the ground truth graph (when available) as a check. We evaluate end-to-end causal inference on all valid combinations that arise from combining *discovery* methods in {DECI-Gaussian (DGa), DECI-

Table 1: Method rank on different (CSuite, Twins, IHDP and ER/SF) datasets, ranking by median ATE RMSE. We present mean $\pm 1$ s.e. of the rank over 27 datasets. Supporting data in Table 3. Bold indicates the possible top methods, accounting for error bars. We treat methods with access to the true graph separately.

| Method | Mean rank |
|---|---|
| DECI Gaussian (DGa) | **6.26 $\pm$ 0.60** |
| DECI Gaussian DoWhy Linear (DGa+L) | 8.37 $\pm$ 0.50 |
| DECI Gaussian DoWhy Nonlinear (DGa+N) | 8.52 $\pm$ 0.51 |
| DECI Spline (DSp) | **6.04 $\pm$ 0.68** |
| DECI Spline DoWhy Linear (DSp+L) | 7.78 $\pm$ 0.60 |
| DECI Spline DoWhy Nonlinear (DSp+N) | **6.63 $\pm$ 0.66** |
| PC + DoWhy Linear (PC+L) | 8.87 $\pm$ 0.41 |
| PC + DoWhy Nonlinear (PC+N) | 7.54 $\pm$ 0.45 |
| True graph DECI Gaussian (T+DGa) | **3.74 $\pm$ 0.47** |
| True graph DECI Spline (T+DSp) | **4.19 $\pm$ 0.56** |
| True graph DoWhy Linear (T+L) | 4.87 $\pm$ 0.58 |
| True graph DoWhy Nonlinear (T+N) | 5.20 $\pm$ 0.71 |

Spline (DSp), PC, and True graph (T)} with causal *inference* methods in {DECI-Gaussian (DGa), DECI-Spline (DSp), DoWhy-Linear (L), DoWhy-Nonlinear (N)}.

**Metrics.** We report RMSE between (C)ATE estimates and the ground truth.

Table 6 provides a high-level summary of our results. For each dataset, we estimated the ATE using each combination of methods, computed the RMSE and took the median over random seeds. We then ranked methods for each dataset (with 1 being the best) and aggregated over the 27 datasets. We find that DECI Spline has the overall best (lowest) rank. RMSE scores are in Appendix E.

In Table 2, we present detailed results for six CSuite datasets. **Lin. Exp** is a two node linear SEM with exponential noise, only DECI Spline can recover the true graph, ATE estimation quality is similar for different estimator once the true graph is found. **Nonlin. Gauss** is a two node non-linear SEM with Gaussian noise, only DECI can fit the highly non-linear functional relationship, with equal performance between DECI-Gaussian and -Spline. **Large backdoor** is a larger nonlinear SEM with non-Gaussian noise in which adjusting for all confounders is valid, but of high variance. For DECI-Spline, which performs well on discovery, the ATE estimation is best using DECI, as DoWhy takes the maximal adjustment set thereby increasing estimator variance.

**Weak arrows** is a similar SEM to Large backdoor, except that a maximal adjustment set is now necessary. Here, DECI-Spline is best for discovery, but is somewhat less accurate for ATE estimation given the right graph. **Nonlin. Simpson** is an adversarially constructed dataset where 1) the true graph is theoretically identifiable, but difficult to discover in practice, 2) ATE estimation is very poor given the wrong graph (Simpson's paradox). All methods perform equally badly. **Symprod Simpson** is a similar but slightly easier dataset, for which DECI-Spline with DML does well.

We performed similar analysis for ATE estimation on ER and SF datasets, on additional CSuite datasets that contain discrete variables or are unidentifiable, and CATE estimation on a subset of CSuite. See Appendix E and Appendix E.4. Also, we test the case with latent confounders in Appendix E.7.

Table 2: Median ATE RMSEs from 20 seeds for six CSuite datasets. Acronyms as in Table 6.

|        | Lin. Exp | Nonlin. Gauss | Large backdoor | Weak arrows | Nonlin. Simpson | Symprod Simpson |
|--------|----------|---------------|----------------|-------------|-----------------|-----------------|
| DGa    | 1.029    | **0.042**     | 0.213          | 1.097       | 1.995           | 0.318           |
| DGa+L  | 1.031    | 1.522         | 0.144          | 1.108       | 1.994           | 0.695           |
| DGa+N  | 1.031    | 1.532         | 0.331          | 1.108       | 1.994           | 0.487           |
| DSp    | 0.022    | **0.043**     | **0.031**      | 0.189       | 1.997           | 0.427           |
| DSp+L  | **0.001** | 1.522        | 0.091          | 0.110       | 1.994           | 0.819           |
| DSp+N  | 0.002    | 1.532         | 0.232          | **0.064**   | 1.994           | **0.160**       |
| PC+L   | 0.516    | 1.532         | 1.690          | 1.108       | 1.994           | 0.487           |
| PC+N   | 0.517    | 1.532         | 1.690          | 1.108       | 1.994           | 0.487           |
| T+DGa  | 0.073    | 0.034         | 0.167          | 0.255       | 0.404           | 0.101           |
| T+DSp  | 0.028    | 0.034         | 0.035          | 0.128       | 0.531           | 0.242           |
| T+L    | 0.001    | 1.522         | 0.105          | 0.109       | 0.848           | 0.819           |
| T+N    | 0.003    | 1.532         | 0.241          | 0.015       | 0.597           | 0.168           |

On the semi-synthetic benchmark datasets, Twins and IHDP, we evaluated both ATE and CATE estimation as shown in Figure 5. For ATE estimation, DECI-Spline is fractionally better than baselines on Twins and significantly better for IHDP. On IHDP, it appears that only DECI-Spline was successful at ATE estimation. Given the right graph (Table 3), DECI-Spline is the best method for computing ATE. For CATE estimation, a similar pattern emerges. Additionaly, DECI-Spline have the similar ATE error as True Graph + DECI Spline (Table 3) for IHDP, implying it is also successful at discovering the graph.

**Summary** Across all experiments we see that DECI enables end-to-end causal inference with competitive performance on both synthetic and more realistic data. DECI particularly performs well compared to other methods when its ability to handle nonlinear functional relationship and non-Gaussian noise distributions comes into play in causal discovery *or* causal inference. Other ECI method combinations can achieve strong performance, but have weak performance if either step's assumptions are violated. We find DECI-Spline particularly attractive given its high degree of flexibility.

# 5 Discussion, scope, and limitations

*Causal inference* requires causal assumptions on the relationships between variables of interest. The field of *causal discovery* aims to learn about these relationships from observational data, given some non-causal assumptions on the data generating process. Motivated by a real-world application where our knowledge of causal relationships is incomplete, DECI combines ideas from causal discovery and inference to go directly from observations to causal predictions. This formulation requires us to adopt assumptions, namely, that the data is generated with a non-linear ANM and that there are no unobserved confounders. Empirically, we find DECI to perform well when these assumptions are satisfied, validating the viability of an end-to-end approach. However, the non-linear ANM assumptions made by DECI are impossible to check in most real-world scenarios. Thus, combining the output of discovery methods with incomplete causal assumptions is an attractive avenue to make end-to-end methods more robust in the future. Interestingly, we also test the scenarios where DECI's assumptions are violated. Specifically under 1. dataset with missingness (section 4.1); 2. dataset with discrete variables (section 4.2); and 3. under latent confounders (appendix E.7), we do not find its performance to degrade severely apart from the latent confounder scenario. This encouraging result motivates us to extend our theoretical analysis to the mixed data type; missing data settings and improving the robustness under latent confounders in future work. Finally, in theoretical considerations, certain important aspects of the end-to-end method remain to be developed, including convergence rate, asymptotic optimality, and dealing with limited overlap and imbalance.

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

# A    Theoretical Considerations for DECI

DECI can be categorized as a functional score-based causal discovery approach, which aims to find the model parameters $\theta$ and mean-field posterior $q_\phi(G)$ by maximizing the ELBO (eq. (9)). A key statistical property of DECI is whether it is capable of recovering the ground truth data generating distribution and true graph $G^0$ when DECI is **correctly specified** and with infinite data. In the following, we will show that DECI is indeed capable of this under standard assumptions. The main idea is to first show that the maximum likelihood estimate (MLE) recovers the ground truth due to the correctly specified model. Then, we prove that optimal solutions from maximizing the ELBO are closely related to the MLE under standard assumptions.

**Theorem 1** (DECI recovers the true data generating process). *Assume DECI's underlying SEM belongs to continuous additive noise structural equation models with (1) non-invertible, $3^{rd}$-order differentiable function between variables that are not constant w.r.t. any of their inputs (i.e. non-trivial); (2) differentiable noise density and bounded log-likelihood, then, under a correctly specified model, infinite data limit and no latent confounders in data generating mechanism, the solution $(\theta', q'_\phi(G))$ from maximizing the ELBO (eq. (9)) satisfies $q'_\phi(G) = \delta(G = G')$ where $G'$ is a unique graph. In particular, $G' = G^0$ and $p_{\theta'}(\mathbf{x}; G') = p(\mathbf{x}; G^0)$.*

Our proof strategy is as follows. In Appendix A.1, we show that the assumptions of the theorem imply certain well-known properties and standard causal assumptions. In Appendix A.2, we prove that maximum likelihood estimation identifies the correct data generating process, and in Appendix A.3, we complete the proof by showing that maximizing the ELBO asymptotically recovers the MLE. Appendix A.4 discusses additional theoretical results for treatment effect estimation.

## A.1    Notation, Assumptions and definitions

First, we define the notation and explain the implication of assumptions required for our theory. We denote a random variable $\mathbf{x} \in \mathbb{R}^D$ with a ground truth data generating distribution $p(\mathbf{x}; G^0)$, where $G^0$ is a binary adjacency matrix representing the true causal DAG. DECI is an additive noise model (ANM), defining the structural equations $x_j = f(\mathbf{x}_{\text{pa}(j;G)}; \theta) + z_j$, where $\text{pa}(j; G)$ are the parents of node $j$ specified by the adjacency matrix $G$, $f$ are non-invertible, third-order differentiable non-trivial functions and $z_j$ are mutually independent noise variables with a joint density $p_\theta(z_1, \ldots, z_D)$. The mean-field variational distribution $q_\phi(G)$ is a product of independent Bernoulli distribution, and $p(G)$ is the soft prior over the graph defined by eq. (5).

For completeness, we first provide the definition of causal minimality, structural identifiability, and explain our model specification and regularity assumptions.

**Definition 1** (Minimality). *For a distribution $p_\theta(\mathbf{x}; G)$ induced by SEM with graph $G$ and parameter $\theta$, we say the model satisfies the causal minimality condition if the distribution $p_\theta(\mathbf{x}; G)$ does not satisfy the local Markov condition with respect to any sub-graph of $G$.*

**Definition 2** (Structural Identifiability). *For a distribution $p_\theta(\mathbf{x}; G)$ induced from SEM, the graph $G$ is said to be structural identifiable from $p_\theta(\mathbf{x}; G)$ if there exists no other distribution $p_{\theta'}(\mathbf{x}; G')$ such that $G \neq G'$ and $p_\theta(\mathbf{x}; G) = p_{\theta'}(\mathbf{x}; G')$.*

The following assumptions are explicit assumptions of Theorem 1, and are restated more formally now, along with a discussion of when such assumptions are likely to hold in practice.

**Assumption 1** (Correctly Specified Model). *We assume the DECI model is correctly specified. Namely, there exists a parameter $\theta^*$ such that $p_{\theta^*}(\mathbf{x}; G^0) = p(\mathbf{x}; G^0)$.*

In practice, this assumption is hard to check in general.

**Assumption 2** (Regularity of log likelihood). *We assume for all parameters $\theta$ and possible graphs $G$, the following holds:*

$$\mathbb{E}_{p(\mathbf{x}; G^0)} \left[ |\log p_\theta(\mathbf{x}; G)| \right] < \infty.$$

This assumption can be verified by analysing the tail behaviour of the noise distribution of the DECI SEM.

We can now show that the our assumptions about the DECI model imply some of the commonly used causal assumptions.

**Lemma 1** (Implication of DECI assumptions). *Assume DECI satisfies the assumptions mentioned in Theorem 1. Then DECI satisfies (1) the* causal Markov assumption*; (2)* causal minimality*; (3)* structural identifiability*.*

*Proof.* We show how each assumptions are implied by referring to some theorems from previous works.

**Causal Markov assumption.** Since DECI belongs to the class of SEMs with differentiable probability density functions, from Theorem 6.22 in Peters et al. (2017), the global, local and Markov factorization properties are equivalent. This also allows the gradient-based optimization to be used for the induced SEM. Theorem 1.4.1 in Pearl (2009c) proves that the induced joint probability from SEM is Markovian with respect to the graph. We typically call this the causal Markov property.

**Causal minimality.** Since we assume the induced distribution from DECI SEM has a density and it is Markovian to its graph, Proposition 6.36 in Peters et al. (2017) proves that if the child node is not independent to any of its parent nodes conditioned on other remaining parents, then it satisfies the causal minimality. Since we assume that the functional relations between output and inputs are not constant, they cannot be independent. Thus, by Proposition 6.36, it satisfies causal minimality.

**Structural identifiability.** From the Proposition 23 in Peters & Bühlmann (2014), the necessary conditions for nonidentifiability require the functions to be either linear or monotonic. However, we assume DECI functional relationship is non-invertible, which violates these conditions. Since we assume causal minimality for DECI SEM, from Theorem 20 in Peters & Bühlmann (2014), DECI SEM is structural identifiable. □

**Comparisons to MEC-based approaches** DECI aims to recover the full causal structures by restricting the functional relationships as ANM. Alternative to this, there exists MEC-based approaches that can identify the causal quantities upto a Markov equivalence class (Guo & Perkovic, 2021; Jung et al., 2020). Guo & Perkovic (2021); Jung et al. (2020) requires the access to the partial ancestral graph (PAG) to start with, and can only work with set-identifiability. For example, Guo & Perkovic (2021) enumerates all possible effects associated to the equivalence class. On the other hand, DECI also enumerates all the causal effects associated to the posterior distribution of the causal graph. One limitation of the above approaches is that they cannot handle latent confounders. In addition, althogh the PAG required by Guo & Perkovic (2021); Jung et al. (2020) can be learned by conditional independence (CI) test, making use of them is non-trivial in practice. Maeda & Shimizu (2020; 2021); Bhattacharya et al. (2021) showed that CI-based approach can underperform the functional-based and differentiable methods, which is the foundations of DECI.

## A.2 MLE Recovers Ground Truth

The likelihood has often been used as the score function for causal discovery. For example, *Carefl* (Khemakhem et al., 2021) adopts the likelihood ratio test (Hyvärinen & Smith, 2013) in the bivariate case, which is equivalent to selecting the causal directions with the maximized likelihood. However, they did not explicitly show that the resulting model recovers the ground truth for the multivariate case. In addition, Zhang et al. (2015) proved that maximizing likelihood for bivariate causal discovery is equivalent to minimizing the dependence between the cause and the noise variable. With the correctly specified, structural identifiable model, the resulting noise and cause are independent through maximizing the likelihood, indicating the graph is indeed causal. However, it is non-trivial to generalize this to the multivariate case that we treat in DECI. In the following, we will show that under a correctly specified model and with maximum likelihood training with infinite data, DECI can recover the unique ground truth graph $G^* = G^0$ and the true data generating distribution $p_{\theta^*}(\mathbf{x}; G^*) = p(\mathbf{x}; G^0)$, where $(\theta^*, G^*)$ are MLE solutions.

**Proposition 1.** *Assuming assumptions in Theorem 1 are satisfied, we denote $(\theta^*, G^*)$ as the MLE solution with infinite training data. Then, we have*

$$p_{\theta^*}(\mathbf{x}; G^*) = p(\mathbf{x}; G^0)$$

*In particular, we have $G^* = G^0$.*

*Proof.* The key idea is to show that with arbitrary $(\theta, G)$, we have the following:

$$\lim_{N \to \infty} \frac{1}{N} \sum_{i=1}^{N} \log p_\theta(\mathbf{x}_i; G) \leq \lim_{N \to \infty} \frac{1}{N} \sum_{i=1}^{N} \log p(\mathbf{x}_i; G^0)$$

By law of large numbers, we have

$$\lim_{N \to \infty} \frac{1}{N} \sum_{i=1}^{N} \log p_\theta(\mathbf{x}_i; G) = \mathbb{E}_{p(\mathbf{x}; G^0)} \left[ \log p_\theta(\mathbf{x}; G) \right].$$

Then, we can show

$$\mathbb{E}_{p(\mathbf{x}; G^0)} \left[ \log p_\theta(\mathbf{x}; G) \right] - \mathbb{E}_{p(\mathbf{x}; G^0)} \left[ \log p(\mathbf{x}; G^0) \right]$$
$$= \mathbb{E}_{p(\mathbf{x}; G^0)} \left[ \log \frac{p_\theta(\mathbf{x}; G)}{p(\mathbf{x}; G^0)} \right]$$
$$\leq \mathbb{E}_{p(\mathbf{x}; G^0)} \left[ \frac{p_\theta(\mathbf{x}; G)}{p(\mathbf{x}; G^0)} - 1 \right] = \int p_\theta(\mathbf{x}; G) d\mathbf{x} - 1 = 0$$

where the inequality is due to $\log t \leq t - 1$. With assumption 3–4, we know there are no latent confounders and the model is correctly specified. Then, the above equality holds when $(\theta^*, G^*)$ induces the same join likelihood $p(\mathbf{x}; G^0)$. Since the model is structural identifiable, we must have $G^* = G^0$. □

## A.3 DECI Recovers the Ground Truth

To show that DECI can indeed recover the ground truth by maximizing the ELBO, we first introduce an important lemma showing the KL regularizer $\frac{1}{N} \mathrm{KL}[q_\phi(G) \| p(G)]$ is negligible in the infinite data limit.

**Lemma 2.** *Assume a variational distribution $q_\phi(G)$ over a space of graphs $\mathcal{G}_\phi$, where each graph $G \in \mathcal{G}_\phi$ has a non-zero associated weight $w_\phi(G)$. With the soft prior $p(G)$ defined as eq. (5) and bounded $\lambda, \rho, \alpha$, we have*

$$\lim_{N \to \infty} \frac{1}{N} \mathrm{KL}[q_\phi(G) \| p(G)] = 0. \tag{15}$$

*Proof.* First, we write down the definition of KL divergence

$$\mathrm{KL}[q_\phi(G) \| p(G)] = \sum_{G \in \mathcal{G}_\phi} w_\phi(G) \left[ \log w_\phi(G) + \lambda \|G\|_F^2 + \rho h(G)^2 + \alpha h(G) + \log Z \right]$$

where $Z$ is the normalizing constant for the soft prior. From the definition and assumptions, it is trivial to know that $\log w_\phi(G)$, $\lambda \|G\|_F^2$ are bounded for all $G \in \mathcal{G}_\phi$. In the following, we show that $h(G)$ and $\log Z$ are also bounded.

From the definition of the DAG penalty, we have $h(G) = \mathrm{tr}(\exp(G \odot G)) - D$. The matrix exponential is defined as

$$\mathrm{tr}(\exp(G \odot G)) = \sum_{k=0}^{\infty} \frac{1}{k!} \mathrm{tr}((G \odot G)^k)$$
$$= \sum_{k=0}^{\infty} \frac{1}{k!} \mathrm{tr}((G)^k)$$
$$= \sum_{k=0}^{D} \frac{1}{k!} \mathrm{tr}((G)^k)$$

where the second equality is due to the fact that $G$ is a binary adjacency matrix. From Zheng et al. (2018b), we know that $\text{tr}(G^k)$ counts for the number of closed loops with length $k$. Since the graph has finite number of nodes, the longest possible closed loop is $D$, resulting in the third equality.

Thus, it is obvious that for any $k$, the number of closed loops with length $k$ must be finite. Hence, it is trivial that $h(G) < \infty$. Therefore, with bounded $\lambda, \rho, \alpha$, the un-normalized soft prior

$$|\exp(-\lambda\|G\|_F^2 - \rho h(G)^2 - \alpha h(G))| < \infty.$$

Thus, the normalizing constant $Z$ must be finite since there are only finite number of possible graphs.

Therefore, these must exists a constant $M_{\phi,G}$ such that $\log w_\phi(G) + \lambda\|G\|_F^2 + \rho h(G)^2 + \alpha h(G) + \log Z < M_{\phi,G}$. Hence, we have

$$0 \le \text{KL}[q_\phi(G)\|p(G)] < \sum_{G\in\mathcal{G}_\phi} w_\phi(G) M_{G,\phi} \le \sqrt{\sum_{G\in\mathcal{G}_\phi} w_\phi^2(G)} \sqrt{\sum_{G\in\mathcal{G}_\phi} M_{G,\phi}^2} < \infty.$$

Thus, we have

$$\lim_{N\to\infty} \frac{1}{N} \text{KL}[q_\phi(G)\|p(G)] = 0$$

where the third inequality is obtained by using Cauchy-Schwarz inequality. $\square$

Now, we can prove Theorem 1.

*Proof.* In terms of optimization, it is equivalent to re-write the ELBO (eq. (9)) as

$$\frac{1}{N}\mathbb{E}_{q_\phi}\left[\log p_\theta(\mathbf{x}_1, \ldots, \mathbf{x}_N | G)\right] - \frac{1}{N}\text{KL}\left[q_\phi(G)\|p(G)\right].$$

Now, under the infinite data limit and the definition of $q_\phi$, we have

$$\lim_{N\to\infty} \frac{1}{N}\mathbb{E}_{q_\phi}\left[\log p_\theta(\mathbf{x}_1, \ldots, \mathbf{x}_N | G)\right] - \frac{1}{N}\text{KL}\left[q_\phi(G)\|p(G)\right]$$

$$= \lim_{N\to\infty} \frac{1}{N} \sum_{G\in\mathcal{G}_\phi} w_\phi(G) \log p_\theta(\mathbf{x}_1, \ldots, \mathbf{x}_N | G) - \frac{1}{N}\text{KL}[q_\phi(G)\|p(G)]$$

$$= \lim_{N\to\infty} \frac{1}{N} \sum_{i=1}^{N} \sum_{G\in\mathcal{G}_\phi} w_\phi(G) \log p_\theta(\mathbf{x}_i | G)$$

$$= \int p(\mathbf{x}; G^0) \sum_{G\in\mathcal{G}_\phi} w_\phi(G) \log p_\theta(\mathbf{x}|G) d\mathbf{x},$$

where the second and third equalities are from Lemma 2 and the law of large numbers, respectively. Let $(\theta^*, G^*)$ be the solutions from MLE (Proposition 1). Then, we have

$$\mathbb{E}_{p(\boldsymbol{x};G^0)}[\log p_\theta(\boldsymbol{x}|G)] \le \mathbb{E}_{p(\boldsymbol{x};G^0)}[\log p_{\theta^*}(\boldsymbol{x};G^*)]$$

for all graph $G \in \mathcal{G}_\phi$ due to the MLE solution $(G^*, \theta^*)$. Then, since $\sum_{G\in\mathcal{G}_\phi} w_\phi(G) = 1$, $w_\phi(G) > 0$, we have

$$\sum_{G\in\mathcal{G}_\phi} w_\phi(G)\mathbb{E}_{p(\mathbf{x};G^0)}\left[\log p_\theta(\mathbf{x}|G)\right] \le \mathbb{E}_{p(\mathbf{x};G^0)}\left[\log p_{\theta^*}(\mathbf{x};G^*)\right]$$

with the equality holding when every graph $G \in \mathcal{G}_\phi$ and associated parameter $\theta_G$ satisfies

$$\mathbb{E}_{p(\mathbf{x};G^0)}\left[\log p_{\theta_G}(\mathbf{x}|G)\right] = \mathbb{E}_{p(\mathbf{x};G^0)}\left[\log p_{\theta^*}(\mathbf{x}|G^*)\right]. \tag{16}$$

Note that when eq. (16) is satified, the ELBO is maximized based on its definition. From proposition 1, under correctly specified model, we have

$$\mathbb{E}_{p(\mathbf{x};G^0)}\left[\log p_{\theta^*}(\mathbf{x}|G^*)\right] = \mathbb{E}_{p(\mathbf{x};G^0)}\left[\log p(\mathbf{x}; G^0)\right]$$

Thus, for a $G' \in \mathcal{G}_\phi$ and associated parameter $\theta'$, the condition in eq. (16) becomes

$$\mathbb{E}_{p(\mathbf{x};G^0)}\left[\log p_{\theta'}(\mathbf{x}|G')\right] = \mathbb{E}_{p(\mathbf{x};G^0)}\left[\log p(\mathbf{x}|G^0)\right]$$
$$\Longrightarrow \mathbb{E}_{p(\mathbf{x};G^0)}\left[\log \frac{p_{\theta'}(\mathbf{x};G')}{p(\mathbf{x};G^0)}\right] = 0$$
$$\Longrightarrow \mathrm{KL}[p(\mathbf{x};G^0)\|p_{\theta'}(\mathbf{x};G')] = 0,$$

which implies $p_{\theta'}(\mathbf{x};G') = p(\mathbf{x};G^0)$. Since DECI is structural identifiable, this means $G' = G^0$ and it is unique. Thus, the graph space $\mathcal{G}_\phi$ only contains one graph $G'$, and $q'_\phi(G) = \delta(G = G')$. $\qquad\square$

One should note that we do not explicitly restrict the noise distribution, indicating it still holds with the spline noise (Section 3.4). However, the above theorem implicitly assumes that DECI is a special case of ANM for structural indentifiability and that the data has no missing values. Thus, it is not applicable for DECI with the mixed-type and missing value extensions. We leave a more general theoretical guarantee to future work.

### A.4 Theoretical considerations for treatment effect estimation

Our proposed DECI model provides an end-to-end framework to connect causal discovery and inference. The assumptions introduced in Appendix A.1 are mainly introduced for the causal discovery task. In the following, we will discuss their relationships to some common assumptions required for the causal inference task. In particular, we show that no additional assumptions are needed to guarantee that DECI can correctly estimate treatment effects.

**Causal effect identifiability and positivity.** Under the assumed DECI SEM (non-linear additive noise model with independent noise variables), our model is Markovian, meaning that the well-known causal effect identification theorem for Markovian models naturally holds (Theorem 3.2.5 of Pearl (2000)). This result does assume positivity of observational distributions.

Fortunately, the proposed form of DECI implies positivity. In particular, DECI is based on non-linear additive noise models. One of the fundamental assumption of non-linear ANMs is that the joint distribution should have a proper density function, i.e. it should be absolutely continuous w.r.t. Lebesgue measure. Additionally, we assume DECI has differentiable density function, so the density is always positive. Additionally, since the noise model for DECI is either a Gaussian or a spline transformed Gaussian, the measure of the set where the density is zero must have a measure zero. This means DECI will have a positive density almost everywhere, which in turn implies positivity.

In a nutshell, one the DECI is learned with assumption 1 and 2, the causal effect associated to the underlying SEM is always identifiable. The remaining question is whether the identified causal effect is the correct estimate of the true effect. This requires the DECI recovers the underlying causal system which is guaranteed by Theorem 1. On potential limitation is that assumption 1,2 and global optimization are hard to verify, meaning that the estimated causal effect may not be the truth.

**A note on consistency assumptions.** The consistency assumption is not required by our method, as these are typically used only within the potential outcome formulation for causality of Rubin (Rubin, 1974), and are rarely seen in graphical model based approach (Pearl, 2000) to causality adopted in our paper. The graphical model approach follows a different philosophy to defined causality: we start by defining a causal model, and all other counterfactual and interventional quantities are derived from this causal model. On the contrary, the PO approach treats counterfactuals (i.e. potential outcomes) as primitive random variables that are not observed. Therefore, the PO approach needs assumptions such as consistency, no interference, etc. to ensure the coherent mathematical consistency of counterfactual statements. See Imbens (2020); Pearl (2009a); Markus (2021), for example, for more discussions on PO approach versus graphical approach to causality.

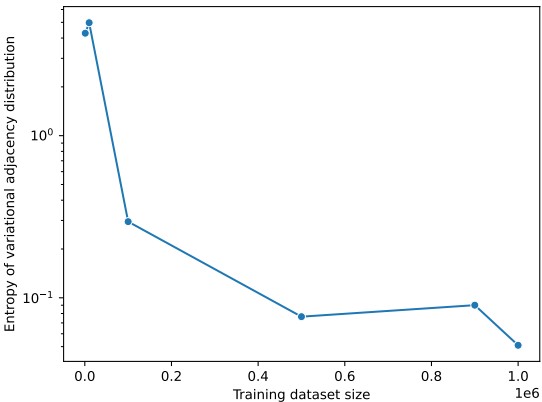

Figure 6: The entropy of the graph posterior $q_\phi$ w.r.t. training dataset size.

**A note on causal graph in causal inference**    For causal inference approaches, like TLearner, SLearner (Künzel et al., 2019), etc., also assume an implicit causal graph. In general, covariates act as confounders, and there is a direct connection from the treatment to the effect variable. However, how to pick the covariate variable is challenging, and most approaches simply pick all remaining variables, leading to an incorrect graph in most cases. Additionally, strong prior knowledge is also required to propose a reasonable graph. For example, it is hard in general to find all possible mediators between treatment and effect. However, DECI relieves this problem by incorporating a graph learning stage. Once trained, DECI can leverage the identified graph distribution to estimate the effect on the target variable. Therefore, only the relevant variables will be selected as covariates, and DECI can naturally deal with mediators.

**Effect to causal estimation of incorrect graph with low probability**    The way DECI used to estimate causal quantities is based on the Monte Carlo approximation due to the intractability of the expectation over graph posterior. Therefore, the quality of the inference depends heavily on the graph distribution and the number of samples used in the MC. If the incorrect graph has very low probability such that it is rarely sampled, then the estimation quality will not be affected with smaller graph sample size. But the variance of the quantity will be high. If the graph sample size increases towards infinity, the estimation variance will tend towards zero, but the incorrect graph will have inneglihible effect, and result in a biased estimator.

## A.5    Convergence to single graph with large dataset size

Based on the theorem 1, the graph posterior tends to collapse to a single graph. We conduct additional experiment to demonstrate this trend. First, we sample a dataset similar to the existing ones from ER with 16 nodes and 64 edges (section 4.1), with $1M$ training data points and 5000 in a separate validation dataset. We then sub-sample the training dataset to contain additional datasets with $1000, 10,000, 100,000, 500,000, 900,000$ points, where each smaller subset is fully contained in each larger one. Finally, we train DECI on those training sets with the same seed, adjusting the augmented Lagrangian parameters linearly with the training set size to ensure a consistent number of training epochs are seen.

From these runs, we plot the entropy of the graph posterior with the training dataset size (Figure 6). We can clearly observe that the entropy decreases with the training data size, indicating its tendency of converging to a single graph.

# B  Additional Details for DECI

## B.1  Optimization Details for Causal Discovery

As mentioned in the main text, we gradually increase the values of $\rho$ and $\alpha$ as optimization proceeds, so that non-DAGs are heavily penalized. Inspired by *Notears*, we do this with a method that resembles the updates used by the augmented Lagrangian procedure for optimization (Nemirovsky, 1999). The optimization process interleaves two steps: (i) Optimize the objective for fixed values of $\rho$ and $\alpha$ for a certain number of steps; and (ii) Update the values of the penalty parameters $\rho$ and $\alpha$. The whole optimization process involves running the sequence (i)–(ii) until convergence, or until the maximum allowed number of optimization steps is reached.

**Step (i).** Optimizing the objective for some fixed values of $\rho$ and $\alpha$ using Adam (Kingma & Ba, 2014). We optimize the objective for a maximum of 6000 steps or until convergence, whichever happens first (we stop early if the loss does not improve for 1500 optimization steps. If so, we move to step (ii)). We use Adam, initialized with a step-size of 0.01. During training, we reduce the step-size by a factor of 10 if the training loss does not improve for 500 steps. We do this a maximum of two times. If we reach the condition a third time, we do not decrease the step-size and assume optimization has converged, and move to step (ii).

**Iterating (i)–(ii).** We initialize $\rho = 1$ and $\alpha = 0$. At the beginning of step (i) we measure the DAG penalty $P_1 = \mathbb{E}_{q_\phi(G)} h(G)$. Then, we run step (i) as explained above. At the beginning of step (ii) we measure the DAG penalty again, $P_2 = \mathbb{E}_{q_\phi(G)} h(G)$. If $P_2 < 0.65\,P_1$, we leave $\rho$ unchanged and update $\alpha \leftarrow \alpha + \rho\,P_2$. Otherwise, if $P_2 \geq 0.65\,P_1$, we leave $\alpha$ unchanged and update $\rho \leftarrow 10\,\rho$. We repeat the sequence (i)–(ii) for a maximum of 100 steps or until convergence (measured as $\alpha$ or $\rho$ reaching some max value which we set to $10^{13}$ for both), whichever happens first.

## B.2  Other Hyperparameters.

We use $\lambda_s = 5$ in our prior over graphs eq. (5). For ELBO MC gradients we use the Gumbel softmax method with a hard forward pass and a soft backward pass with temperature of 0.25.

The functions eq. (7) used in DECI's SEM, $\zeta$ and $\ell$, are 2 hidden layer MLPs with 128 hidden units per hidden layer. These MLPs use residual connections and layer-norm at every hidden layer.

For the non-Gaussian noise model in eq. (8), the bijection $\kappa$ is an 8 bin rational quadratic spline (Durkan et al., 2019) with learnt parameters.

In section 3.3, for ATE estimation we compute expectations by drawing 1000 graphs from DECI's graph posterior $q_\phi$ and for each graph we draw 2 samples of $\mathbf{x}_Y$ for a total of 2000 samples. For CATE estimation, we need to train a separate surrogate predictor per graph samples. We draw 10 different graph samples and 10000 $(\mathbf{x}_C, \mathbf{x}_Y)$ pair samples for each graph. We use these to train the surrogate models.

Our surrogate predictor is a basis function linear model with 3000 random Fourier features drawn such that the model approximates a Gaussian process with a radial basis function kernel of lengthscale equal to 1 (Yu et al., 2016).

## B.3  ELBO Derivation

The goal of maximum likelihood involves maximizing the likelihood of the observed variables. For DECI (with fully observed datasets) this corresponds to the log-marginal likelihood

$$\log p_\theta(x^1, \dots, x^N) = \log \sum_A p(G) \prod_n p_\theta(x^n | G). \tag{17}$$

Marginalising $G$ in the equation above is intractable, even for moderately low dimensions, since the number of terms in the sum grows exponentially with the size of $G$ (which grows quadratically with the data dimensionality $D$).

Variational inference proposes to use a distribution $q_\phi(G)$ to build the ELBO, a lower bound of the objective from eq. (17), as follows:

$$\log p_\theta(x^1, \ldots, x^N) = \log \sum_G p(G) \prod_n p_\theta(x^n|G) \tag{18}$$

$$= \log \sum_G q_\phi(G) \frac{p(G) \prod_n p_\theta(x^n|G)}{q_\phi(G)} \tag{19}$$

$$= \log \mathbb{E}_{q_\phi(G)} \left[ \frac{p(G) \prod_n p_\theta(x^n|G)}{q_\phi(G)} \right] \tag{20}$$

$$\geq \mathbb{E}_{q_\phi(G)} \left[ \log \frac{p(G) \prod_n p_\theta(x^n|G)}{q_\phi(G)} \right] \qquad \text{(Jensen's inequality)} \tag{21}$$

$$= \mathbb{E}_{q_\phi(G)} \left[ \log p(G) \prod_n p_\theta(x^n|G) \right] + H(q_\phi) \tag{22}$$

$$= \mathrm{ELBO}(\phi, \theta), \tag{23}$$

where we denote $H(q_\phi) = -\mathbb{E}_{q_\phi(G)} \log q_\phi(G)$ for the entropy of the distribution $q_\phi$. Interestingly, the distribution $q_\phi$ that maximizes the ELBO is exactly the one that minimizes the KL-divergence between the approximation and the true posterior, $\mathrm{KL}(q_\phi(G)\|p_\theta(G|x^1 \ldots, x^N))$ (see, e.g. Blei et al. (2017)). This is why $q_\phi$ can be used as a posterior approximation.

## B.4 Intervened Density Estimation with DECI

Apart from (C)ATE estimation, DECI may also be used to evaluate densities under intervened distributions. For a given graph, the density of some observation vector $\mathbf{a}$ is computed by evaluating the base distribution density after inverting the SEM

$$p_\theta(\mathbf{x} = \mathbf{a}|G^m) = \prod_i p(\mathbf{z}_i = (\mathbf{a}_i - f_i(\mathbf{a}_{\mathrm{pa}(i;G^m)}))) \tag{24}$$

noting that the transformation Jacobian is the identity. We then marginalise the graphs using Monte Carlo:

$$p_\theta(\mathbf{x} = \mathbf{a}) \approx \frac{1}{M} \sum_m^M p_\theta(\mathbf{x} = \mathbf{a}|G^m); \quad G^m \sim q_\phi(G). \tag{25}$$

In the rest of this section we derive methods that allow using DECI to estimate causal quantities.

Under $G_{\mathrm{do}(\mathbf{x}_T)}$, $i \in T$ correspond to parent nodes and we have the following factorisation: $p(\mathbf{x}|G_{\mathrm{do}(\mathbf{x}_T)}) = p(\mathbf{x}_{\backslash T}|G_{\mathrm{do}(\mathbf{x}_T)}) \prod_{i \in T} p(\mathbf{x}_i)$. We can then evaluate the interventional density of an observation $\mathbf{x}_{\backslash T} = \mathbf{a}$ with DECI as

$$
\begin{aligned}
& p_\theta(\mathbf{x}_{\backslash T} = \mathbf{a}|\mathrm{do}(\mathbf{x}_T = \mathbf{b}), G^m) \\
& = \frac{p_\theta(\mathbf{x}_{\backslash T} = \mathbf{a}, \mathbf{x}_T = \mathbf{b}|G^m_{\mathrm{do}(\mathbf{x}_T)})}{p_\theta(\mathbf{x}_T = \mathbf{b}|G^m_{\mathrm{do}(\mathbf{x}_T)})} \\
& = \frac{p_\theta(\mathbf{x}_{\backslash T} = \mathbf{a}|\mathbf{x}_T = \mathbf{b}, G^m_{\mathrm{do}(\mathbf{x}_T)}) p_\theta(\mathbf{x}_T = \mathbf{b})}{p_\theta(\mathbf{x}_T = \mathbf{b})} \\
& = \prod_{j \in \backslash T} p(\mathbf{z}_i = (\mathbf{a}_i - f_i(\mathbf{a}_{\mathrm{pa}(i;G^m_{\mathrm{do}(\mathbf{x}_T)})}))),
\end{aligned} \tag{26}
$$

which amounts to evaluating the density of the exogenous noise correspondint to non-intervened variables. We can then marginalise the graph using Monte Carlo as in eq. (25).

### B.5 Relationship with Khemakhem et al. (2021)

Khemakhem et al. (2021) introduced *Carefl*, a method that uses autoregressive flows (Huang et al., 2018b; Kingma et al., 2016) to learn causal-aware models, using the variables' causal ordering to define the autoregressive transformations. The method's main benefit is its ability to model complex nonlinear relationships between variables. However, *Carefl* alone is insufficient for causal discovery, as it requires the causal graph structure as an input. The authors propose a two-step approach. First, run a traditional constraint-based method (e.g., PC) to find the graph's skeleton and orient as many edges as possible, and second, fit several flow models to determine the orientation of the remaining edges. The drawbacks of this approach include the dependence on an external causal discovery methods (which will inherently limit *Carefl*'s performance to that of the method used), and the cost of fitting multiple flow models to orient the edges that are left unoriented after the first step. Our method extends Khemakhem et al. (2021) to learn the causal graph among multiple variables and perform end-to-end causal inference.

### B.6 Discussion on Causal Discovery Methods

When performing causal discovery, DECI returns a posterior over graphs. Most other causal discovery methods return either a single graph or an equivalence class of graphs. However, we can re-cast these methods in the probabilistic framework used by DECI by noting that a posterior over graphs takes the form

$$p(G|\mathbf{X}) = \frac{p(\mathbf{X}|G)p(G)}{\sum_G p(\mathbf{X}|G)p(G)}. \tag{27}$$

In this equation, the likelihood measures the degree of compatibility of a certain DAG architecture with the observed data. For score-based discovery methods (Chickering, 2002; Chickering & Meek, 2015; Chickering, 2020; Huang et al., 2018a) we take the score to be $\log p(\mathbf{X}|G)$. For functional discovery methods (Hoyer et al., 2008b; Shimizu et al., 2006; ZHANG, 2009) we use the exogenous variable log-density. Constraint-based methods (Spirtes & Glymour, 1991; Spirtes et al., 2000) can also be cast in this light by assuming a uniform distribution over all graphs in their outputted equivalence class $\mathcal{G}$: $\log p(\mathbf{X}|G) = -\log|\mathcal{G}|, \forall G \in \mathcal{G}$. To what degree these methods succeed at constraining the space of possible graphs will depend on how well their respective assumptions are met and the amount of data available (Hoyer et al., 2008a).

### B.7 Learning from interventional dataset

In this section, we discuss how to extend our method to enable learning from interventional dataset. Assuming we have gathered $S$ different interventional datasets $\mathcal{D}_s = \{\mathbf{x}_{(s)}^1, \ldots, \mathbf{x}_{(s)}^{N_s}\}, s = 1, \ldots, S$ from the *same* causal system; and for each $\mathcal{D}_s$, it is associated with a set of indices of interventional variables, $\tau_{(s)}$. For pure observational dataset, we assume $\tau_{(s)} =$. Then, the training objective becomes:

$$\text{ELBO}(\theta, \phi) = \sum_s \mathbb{E}_{q_\phi(G)} \left[ \log p(G) \prod_{n_s} p_\theta(\mathbf{x}_{(s)}^{n_s} | G_{\text{do}(\mathbf{x}_{\tau_{(s)}})}) \right] + H(q_\phi) \tag{28}$$

where $G_{\text{do}(\mathbf{x}_{\tau_{(s)}} = \mathbf{x}_{\tau_{(s)}}^{n_s})}$ denotes the mutilated graph under intervention $\mathbf{x}_{\tau_{(s)}}$. When $S = 1$ and $\tau_{(1)} =$, Equation (28) reduces to the fully observational case, Equation (9).

## C Unified View of Causal Discovery Methods

This section introduces a simple analysis showing that, similarly to DECI, most causal discovery methods based on continuous optimization can be framed from a probabilistic perspective as fitting a flow. The benefits of this unified perspective are twofold. First, it allows a simple comparison between methods, shedding light on the different assumptions used by each one, their benefits and drawbacks. Second, it simplifies the development of new tools to improve these methods, since any improvements to one of them can be easily mapped to the others by framing them in this unified framework (e.g. our extensions to handle missing values and flexible noise distributions can be easily integrated with *Notears*).

The connection between causal discovery methods based on continuous optimization and flow-based models uses the concept of a weighted adjacency matrix $W(\theta) \in \mathbb{R}^{D \times D}$ linked to a function $f(\mathbf{x}; \theta) : \mathbb{R}^D \to \mathbb{R}^D$. Loosely speaking, these matrices can be seen as characterizing how likely is each output of $f(\mathbf{x}; \theta)$ to depend on each component of the input $\mathbf{x}$. For instance, $W(\theta)_{j,i} = 0$ indicates that $f_i(\mathbf{x}; \theta)$ is completely independent of $x_j$. Such adjacency matrices can be constructed efficiently for a wide range of parameterizations for $f$, such as multi layer perceptrons and weighted combinations of nonlinear functions. We refer the reader to Zheng et al. (2020) for details. In the following, lemma 4 shows that the objective function of fitting the flow is equivalent to solving the maximum likelihood problem of the noise distribution. Therefore, this allows the objective functions in many existing causal discovery methods to be unified under the maximum likelihood perspective.

**Lemma 3.** *Let $G$ represents a binary adjacency matrix, $f_G : \mathbb{R}^d \to \mathbb{R}^d$ a function whose $i^{th}$ output only depends on the parents of $x_i$ specified in $G$, and $J_G(\mathbf{x})$ be the Jacobian of $g_G(\mathbf{x}) = \mathbf{x} - f_G(\mathbf{x})$. If $G$ represents a DAG, then $|\det J_G(\mathbf{x})| = 1$*

*Proof.* We split the proof in several simple steps.

1. $g_G(\mathbf{x})$ has a Jacobian-determinant of $\det(I - J_f)$, where $J_f$ is the Jacobian of $f_G(\mathbf{x})$.

2. $J_f$ has non-zero entries exactly in the positions where $G^T$ is non-zero. Therefore, it remains the DAG structure.

3. Matrix with DAG structure is nilpotent (i.e. all eigenvalues are zero). Thus, $J_f$ can be factorized into $QUQ^*$ where $Q$ is unitary and $U$ is strictly upper triangular (Schur factorization).

4. Finally, $\det(I - J_f) = \det(I - QUQ^*) = \det(I - U) = 1$

$\square$

**Lemma 4.** *Let $f(\mathbf{x}; \theta) : \mathbb{R}^D \to \mathbb{R}^D$ be a $\theta$-parameterized function with weighted adjacency matrix $W(\theta) \in \mathbb{R}^{D \times D}$. Given a dataset $\{\mathbf{x}^1, \dots, \mathbf{x}^N\}$, fitting a flow with the transformation $\mathbf{z} = \mathbf{x} - f(\mathbf{x}; \theta)$, base distribution $p_\mathbf{z}$ and a hard acyclicity constraint on $W(\theta)$ is equivalent to solving*

$$\max_\theta \sum_{n=1}^{N} \log p_\mathbf{z}(\mathbf{x}^n - f(\mathbf{x}^n; \theta)) \quad \text{s.t.} \quad h(W(\theta)) = 0, \tag{29}$$

*where $h(\cdot)$ is the algebraic characterization of DAGs from (Zheng et al., 2018a).*

*Proof.* The acyclicity constraint is enforced by constraining the optimization domain to $\Theta = \{\theta : h(W(\theta)) = 0\}$. Then, the maximum likelihood objective can be written as

$$\sum_n \log p_\theta(\mathbf{x}^n) = \sum_n \log p_z(\mathbf{x}^n - f(\mathbf{x}^n; \theta)) + \log \left| \det \frac{\mathrm{d}(\mathbf{x}^n - f(\mathbf{x}^n; \theta))}{\mathrm{d}\mathbf{x}^n} \right| \tag{30}$$

$$= \sum_n \log p_z(\mathbf{x}^n - f(\mathbf{x}^n; \theta)), \tag{31}$$

where the first equality we use the change of variable formula, valid because the transformation $\mathbf{z} = g(\mathbf{x}; \theta) = \mathbf{x} - f(\mathbf{x}; \theta)$ is invertible for any $\theta \in \Theta$, and the second equality uses that the function $x^n - f(\mathbf{x}^n; \theta)$ has Jacobian-determinant equal to 1 (lemma 3), due to the constraint $\Theta = \{\theta : h(W(\theta)) = 0\}$. $\square$

In summary, recently proposed causal discovery methods based on continuous optimization can be formulated from a probabilistic perspective as fitting a flow with different constraints, transformations, and base distributions. This unified formulation sheds light on the assumptions done by each method (e.g. a Gaussian noise assumption, either implicitly as in *Notears* or explicitly as in *Grandag*) and, more importantly, simplifies the development of new tools to improve them. For instance, the ideas proposed to deal with partially-observed datasets and non-Gaussian noise are readily applicable to any of the causal discovery methods mentioned in this section, addressing some of their limitations (Kaiser & Sipos, 2021; Loh & Bühlmann, 2014; Reisach et al., 2021).

# D   Datasets Details

Our two benchmark datasets are constructed following similar procedures described in Louizos et al. (2017).

**IHDP (Hill, 2011).** This dataset contains measurements of both infants (birth weight, head circumference, etc.) and their mother (smoked cigarettes, drank alcohol, took drugs, etc) during real-life data collected in a randomized experiment. The main task is to estimate the effect of home visits by specialists on future cognitive test scores of infants. The outcomes of treatments are simulated artificially as in (Hill, 2011); hence the outcomes of both treatments (home visits or not) on each subject are known. Note that for each subject, our models are only exposed to only one of the treatments; the outcomes of the other potential/counterfactual outcomes are hidden from the mode, and are only used for the purpose of ATE/CATE evaluation. To make the task more challenging, additional confoundings are manually introduced by removing a subset (non-white mothers) of the treated children population. In this way we can construct the IHDP dataset of 747 individuals with 6 continuous covariates and 19 binary covariates. We use 10 replicates of different simulations based on setting B (log-linear response surfaces) of (Hill, 2011), which can downloaded from `https://github.com/AMLab-Amsterdam/CEVAE`. We use a 70%/30% train-test split ratio. Before training our models, all continuous covariates are normalized.

**TWINS (Almond et al., 2005).** This dataset consists of twin births in the US between 1989-1991. Only twins which with the same sex born weighing less than 2kg are considered. The treatment is defined as being born as the heavier one in each twins pair, and the outcome is defined as the mortality of each twins in their first year of life. Therefore, by definition for each pair of twins, we can observe the outcomes of both treatments (the lighter twin and heavier twin). However, during training, only one of the treatment is visible to our models, and the other potential outcome is unknown to the model and are only used for evaluation. The raw dataset is downloaded from `https://github.com/AMLab-Amsterdam/CEVAE`. Following Louizos et al. (2017), we also introduce artificial confounding using the categorical `GESTAT10` variable. This is done by assigning treatments (factuals) using the conditional probability $t_i|\mathbf{x}_i, z_i = \text{Bern}(\sigma(w_0^T \mathbf{x}_i + w_h(z_i/10 - 0.1)))$, where $t_i$ is the treatment assignment for subject $i$, $z_i$ is the corresponding `GESTAT10` covariate, $\mathbf{x}_i$ denotes the other remaining covariates. Both $w_0$ and $w_h$ are randomly generated as $w_0 \sim \mathcal{N}(0, 0.1 * I)$, $w_h \sim \mathcal{N}(5, 0.1)$. All continuous covariates are normalized.

**Ground Truth ATE and CATE Estimation for TWINS and IHDP.** In both benchmark datasets, since the held-out hypothetical outcomes of counterfactual treatments are already known, the the ground truth ATE can be naively estimated by averaging the difference between the factual and counterfactual outcomes across the entire dataset. The CATE estimation is a bit tricky, since both datasets contains covariates collected from real-world experiments, in which the underlying ground truth causal graph structure is unknown. As a result, exact CATE estimation is generally impossible for continuous conditioning sets. Therefore, when evaluating the CATE estimation performance on **TWINS** and **IHDP**, we focus only on discrete variables (binary and categorical) as conditioning set. This allows unbiased estimation of ground truth CATE by simply averaging the treatment effects on subgroups of subjects in the dataset, that have the corresponding discrete value in the conditioning set. We consider only single conditioning variable at a time, and estimate the corresponding CATE for evaluation.

## D.1   CSuite

We develop Causal Suite (CSuite), a number of small to medium (2–12 nodes) synthetic datasets generated from hand-crafted Bayesian networks with the intention of testing different capabilities of causal discovery and inference methods. All continuous-only datasets take the form of additive noise models.

Each dataset comes with a training set of 2000 samples, and between 1 and 2 intervention test sets. Each intervention test set has a treatment variable, treatment value, reference treatment value and effect variable. We estimate the ground truth ATE by drawing 2000 samples from the treated and reference intervened distributions. For the datasets used to evaluate CATE, we generate samples from *conditional* intervened distributions by using Hamiltonian Monte Carlo. We employ a burn-in of 10k steps and a thinning factor of 5 to generate 2000 conditional samples, which we then use to compute our ground truth CATE estimate. We note that because all ground truth causal quantities are estimated from samples, there is a lower bound

on the expected error that can be obtained by our methods. When methods obtain an error equal or lower we say that they have solved the task.

**lingauss**  A two node graph (Figure 7a) with a linear relationship and Gaussian noise. We have $X_1 \sim N(0,1)$ and $X_2 = \frac{1}{2}X_1 + \frac{\sqrt{3}}{2}Z_2$ where $Z_2 \sim N(0,1)$ is independent of $X_1$. The observational distribution is symmetrical in $X_1 \leftrightarrow X_2$. The graph is not identifiable. The best achievable performance on this dataset is obtained when there is a uniform distribution over edge direction.

**linexp**  A two node graph (Figure 7a) with a linear functional relationship, but with exponentially distributed additive noise. We have $X_1 \sim N(0,1)$ and $X_2 = \frac{1}{2}X_1 + \frac{\sqrt{3}}{2}(Z_2 - 1)$ where $Z_2 \sim \text{Exp}(1)$ is independent of $X_1$. By using non-Gaussian noise, the graph becomes identifiable. However, the inference problem will be more challenging for methods sensitive to outliers, such as those that assume Gaussian noise.

**nonlingauss**  A two node graph (Figure 7a) with a nonlinear relationship and Gaussian additive noise. We have $X_1 \sim N(0,1)$ and $X_2 = \sqrt{6}\exp(-X_1^2) + \alpha Z_2$ where $Z_2 \sim N(0,1)$ is independent of $X_1$ and $\alpha^2 = 1 - 6\left(\frac{1}{\sqrt{5}} - \frac{1}{3}\right)$. Note $\text{Var}(X_2) = 1$ and $\text{Cov}(X_1, X_2) = 0$. By having a linear correlation of zero between $X_1$ and $X_2$, this dataset creates a potential failure mode for causal inference methods that assume linearity.

**nonlin_simpson**  A synthetic Simpson's paradox, using the graph Figure 7b: if the confounding factor $X_3$ is not adjusted for, the relationship between the treatment $X_1$ and effect $X_2$ reverses. The variable $X_4$ correlates strongly with the effect, but must not be used for adjustment. Choosing an incorrect adjustment set when estimating $\mathbb{E}[X_2|\text{do}(X_1)]$ leads to a significantly incorrect ATE estimate. All variables are continuous, with nonlinear structural equations and non-Gaussian additive noise.

**symprod_simpson**  Another Simpson's paradox using the graph Figure 7c. This dataset is similar to nonlin_simpson with 2 key differences: 1) the effect variable is the result of a product between the confounding variable and the treatment variable. This makes drawing causal inferences require non-linear function estimation. Additionally, the ATE is close to 0. The conditioning variable for the CATE task is a descendant of the confounding variable. This dataset probes for methods' capacity to reduce their uncertainty about a confounding variables based on values of its child variables.

**large_backdoor**  A nine node graph, as shown in Figure 7d. This dataset is constructed so that there are many possible choices of backdoor adjustment set. While both minimal and maximal adjustment sets can result in a correct solution, the a minimal adjustment set results in a much lower-dimensional adjustment problem and thus will result in lower variance solutions. The conditioning node for the CATE task is a child of the root variable. Thus the CATE task probes for methods' capacity to infer the value of an observed confounder from one of its children. All variables are continuous, with nonlinear structural equations and non-Gaussian additive noise.

**weak_arrows**  A nine node graph, as shown in Figure 7e. Unlike the previous dataset, when the true graph is known, a large adjustment set must be used. The causal discovery challenge revolves around finding all arrows, which are scaled to be relatively weak, but which have significant predictive power for $X_9$ in aggregate. This dataset tests methods' capacity to identify the full adjustment set and adjust for a large number of variables simultaneously.

**cat_to_cts**  A two node (Figure 7a) graph with categorical $X_1$ and continuous $X_2$ with an additive noise model. We have $X_1 \sim \text{Cat}\left(\frac{1}{4}, \frac{1}{4}, \frac{1}{2}\right)$ takes values in $\{0, 1, 2\}$ and $X_2 = X_1 + \frac{8}{5}(s(Z_2) - 1)$ where $s(x) = \log(\exp(x) + 1)$ is the softplus function, and $Z_2 \sim N(0,1)$ is independent of $X_1$.

**cts__to__cat**  A two node (Figure 7a) graph with continuous $X_1$ and categorical $X_2$. We take $X_1 \sim U(-\sqrt{3}, \sqrt{3})$ and $X_2$ categorical on $\{0, 1, 2\}$ with the following conditional probabilities

$$p(X_2|X_1 = x_1) = \begin{cases} \left(\frac{6}{13}, \frac{6}{13}, \frac{1}{13}\right) & \text{if } x_1 < -\frac{\sqrt{3}}{3} \\ \left(\frac{1}{8}, \frac{3}{4}, \frac{1}{8}\right) & \text{if } -\frac{\sqrt{3}}{3} \le x_1 < \frac{\sqrt{3}}{3} \\ \left(\frac{1}{3}, \frac{1}{3}, \frac{1}{3}\right) & \text{if } x_1 > \frac{\sqrt{3}}{3} \end{cases} \tag{32}$$

In this problem, we treat $X_2$ as the treatment and $X_1$ as the target, giving a theoretical ATE of zero.

**mixed__simpson**  Similar to the nonlin_simpson dataset, using the graph of Figure 7b, but with $X_3$ categorical on three categories, and $X_1$ binary.

**large__backdoor__binary__t**  Similar to the large_backdoor dataset, using the graph of Figure 7d, but with $X_8$ binary.

**weak__arrows__binary__t**  Similar to the weak_arrows dataset, using the graph of Figure 7e, but with $X_8$ binary.

**mixed__confounding**  A large, mixed type dataset with 12 variables, as shown in Figure 7f. In this dataset, $X_1, X_5$ are binary, $X_3, X_6, X_8$ are categorical on three categories, and other variables are continuous. We utilise nonlinear structural equations and non-Gaussian additive noise.

# E   Additional Results

## E.1   Causal Discovery Results under Gaussian Exogenous Noise

Figure 4 in the main text shows causal discovery results for the case where synthetic data was generated using non-Gaussian noise. In that case it was observed that using DECI together with a flexible noise model performed better than DECI with a Gaussian noise model. Figure 8 shows results for synthetic data generated using Gaussian noise. As expected, in this case using a Gaussian noise model is beneficial, although DECI with a spline noise mode still performs strongly.

## E.2   Summary of CSuite Results

Comprehensive results on CSuite ATE and CATE performance are shown in figures 9 and 10. We first provide a summary of results here and then go into per-dataset analysis in the following subsection.

We find DECI to perform consistently well in our 2 node datasets. It learns a uniform posterior over graphs in the non-identifiable setting, it fits non-linear functions well and it is robust to heavy tailed noise when employing the spline noise model. We find linear and non-linear DML inference to also perform acceptably, with the exception of the heavy tailed noise case, where the methods overfit to outliers and thus estimate ATE poorly.

On the larger (4 and 12 node) datasets, when the true graph is available, DECI provides ATE estimates competitive with the well-established non-linear DML method. Notably, DECI outperforms backdoor adjustment methods when the number of possible adjustment sets is large. Choosing the optimal adjustment set is an np-hard problem and the most common approach is to simply choose the largest one. This leads to doWhy suffering from variance. DECI's simulation-based approach avoids having to choose an adjustment set. On the other hand, for densely connected graphs where the strength of the connection between nodes is low, DECI struggles to capture the funcitonal relationships in the data and DML is most competitive. For CATE estimation DECI provides superior performance in all datasets and is able to completely solve all tasks but one.

When the graph is learnt from the data, the non-linear nature of our (4 and 12 node) datasets together with their heavy tailed noise make the discovery problem very challenging. We find that the PC algorithm

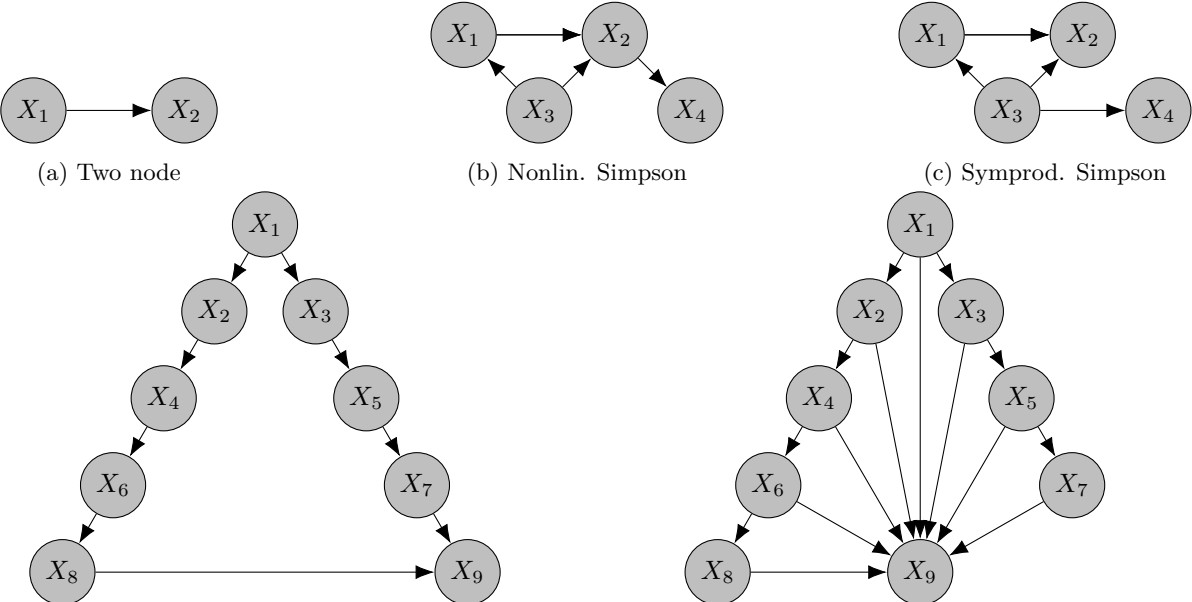

(a) Two node

(b) Nonlin. Simpson

(c) Symprod. Simpson

(d) Large backdoor. Treatment $X_8$, outcome $X_9$, condition $X_7$.

(e) Weak arrows. Treatment $X_8$, outcome $X_9$, condition $X_7$.

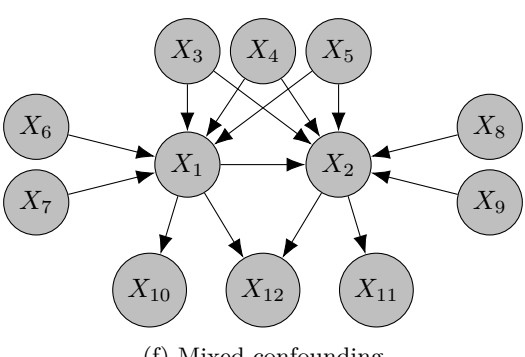

(f) Mixed confounding

Figure 7: CSuite graphs. Unless otherwise stated, we take $X_1$ as the treatment, $X_2$ as the outcome, and for CATE we take $X_3$ as the conditioning variable.

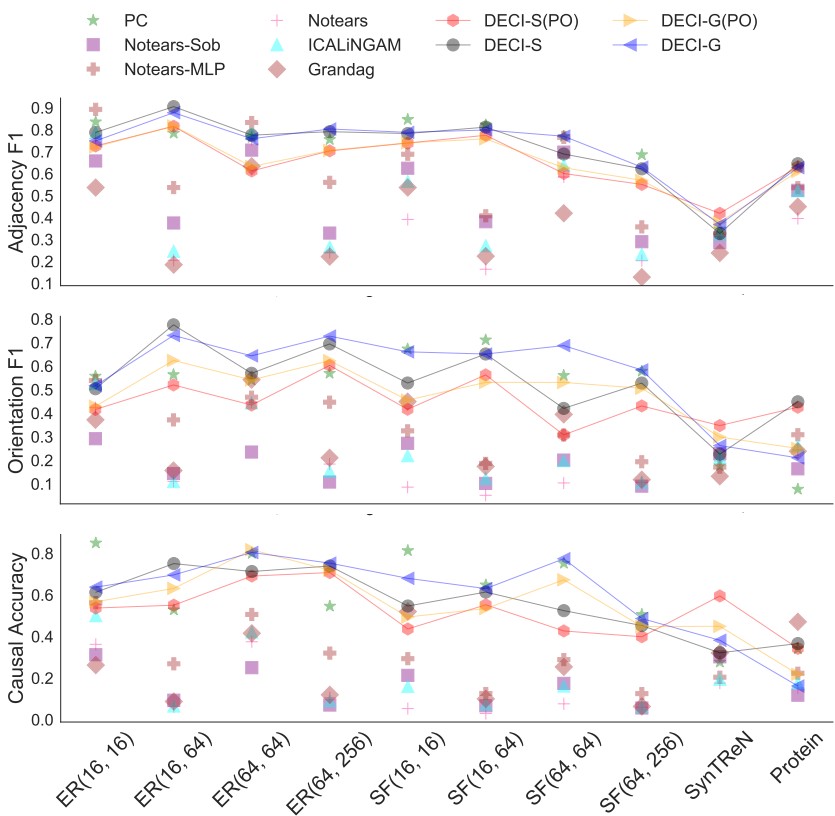

Figure 8: **DECI achieves better results than the baselines in all metrics shown.** The plots show the results for causal discovery for synthetic data generated using Gaussian noise. The legend "DECI-G" and "DECI-S" correspond to DECI using a Gaussian and spline noise model. Additionally, the "(PO)" corresponds to running DECI with 30% of the training data missing completely at random. For readability, we highlight the DECI results by connecting them with soft lines. The figure shows mean results across five different random seeds.

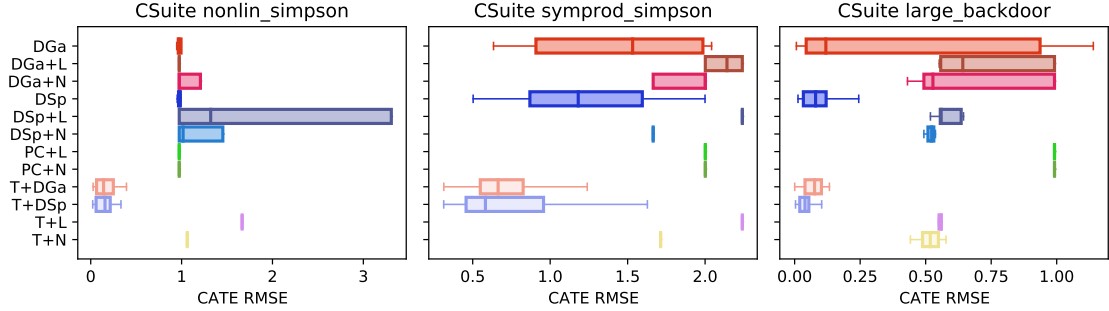

Figure 9: End-to-end ATE results on CSuite.

Figure 10: End-to-end CATE results on CSuite. Colours and acronyms as in Figure 9.

| Method / Dataset | DGa | DGa+L | DGa+N | DSp | DSp+L | DSp+N | PC+L | PC+N | T+L | T+N | T+DGa | T+DSp |
|---|---|---|---|---|---|---|---|---|---|---|---|---|
| ER(16, 16) - G | 1.280 | 1.141 | 1.129 | 1.648 | 1.374 | 1.393 | 1.491 | 1.475 | 0.945 | 0.936 | 1.083 | 1.332 |
| ER(16, 16) - S | 1.726 | 1.776 | 1.780 | 1.829 | 1.776 | 1.755 | 1.869 | 1.790 | 1.755 | 1.793 | 1.643 | 1.660 |
| ER(16, 64) - G | 1.699 | 1.422 | 1.334 | 1.501 | 1.442 | 1.202 | 1.335 | 1.440 | 1.310 | 1.369 | 1.460 | 1.644 |
| ER(16, 64) - S | 2.311 | 2.465 | 2.600 | 2.174 | 2.452 | 2.421 | 2.584 | 2.276 | 2.742 | 2.641 | 2.510 | 2.428 |
| ER(64, 64) - G | 1.208 | 1.420 | 1.190 | 1.287 | 1.450 | 1.397 | 1.325 | 1.273 | 1.284 | 1.250 | 1.158 | 1.124 |
| ER(64, 64) - S | 2.246 | 1.626 | 1.626 | 1.892 | 2.325 | 2.292 | 1.526 | 2.446 | 2.442 | 2.441 | 2.490 | 2.481 |
| SF(16, 16) - G | 1.156 | 1.030 | 1.409 | 1.699 | 2.052 | 1.343 | 1.574 | 1.233 | 1.131 | 1.078 | 1.127 | 1.375 |
| SF(16, 16) - S | 2.870 | 1.805 | 2.284 | 2.431 | 2.363 | 1.805 | 3.008 | 2.520 | 2.518 | 2.502 | 2.424 | 2.477 |
| SF(16, 64) - G | 1.702 | - | - | 1.539 | - | 1.635 | 1.464 | 1.551 | 1.510 | 1.463 | 1.559 | 1.486 |
| SF(16, 64) - S | 3.594 | - | - | 4.139 | - | - | 3.877 | 4.145 | 4.162 | 4.106 | 4.161 | 3.861 |
| SF(64, 64) - G | 1.049 | 0.998 | 1.134 | 1.010 | 1.035 | 1.309 | 0.972 | 1.343 | 1.006 | - | 1.087 | 1.288 |
| SF(64, 64) - S | 2.754 | 3.239 | 3.239 | 3.242 | 3.239 | 3.239 | 3.227 | 2.591 | 2.815 | - | 2.883 | 3.034 |
| csuite_cat_to_cts | 0.495 | 0.504 | 0.504 | 0.501 | 0.504 | 0.504 | 0.262 | 0.264 | 0.020 | 0.023 | 0.019 | 0.036 |
| csuite_cts_to_cat | 0.015 | 0.011 | 0.011 | 0.016 | 0.011 | 0.011 | 0.039 | 0.038 | 0.011 | 0.011 | 0.011 | 0.011 |
| csuite_large_backdoor | 0.213 | 0.144 | 0.331 | 0.031 | 0.091 | 0.232 | 1.690 | 1.690 | 0.105 | 0.241 | 0.167 | 0.035 |
| csuite_large_backdoor_bt | 0.028 | 0.029 | 0.187 | 0.029 | 0.029 | 0.027 | 0.039 | 0.039 | 0.119 | 0.070 | 0.021 | 0.014 |
| csuite_linexp | 1.029 | 1.031 | 1.031 | 0.022 | 0.001 | 0.002 | 0.516 | 0.517 | 0.001 | 0.003 | 0.073 | 0.028 |
| csuite_lingauss | 0.120 | 0.523 | 0.024 | 0.149 | 0.025 | 0.024 | 0.498 | 0.498 | 0.025 | 0.022 | 0.076 | 0.085 |
| csuite_mixed_confounding | 0.477 | 0.461 | 0.282 | 0.254 | 0.500 | 0.380 | 0.250 | 0.387 | 0.107 | 0.057 | 0.019 | 0.018 |
| csuite_mixed_simpson | 1.259 | 0.723 | 0.723 | 1.765 | 1.772 | 1.771 | 0.723 | 0.723 | 0.017 | 0.022 | 0.014 | 0.013 |
| csuite_nonlin_simpson | 1.995 | 1.994 | 1.994 | 1.997 | 1.994 | 1.994 | 1.994 | 1.994 | 0.848 | 0.597 | 0.404 | 0.531 |
| csuite_nonlingauss | 0.042 | 1.522 | 1.532 | 0.043 | 1.522 | 1.532 | 1.532 | 1.532 | 1.522 | 1.532 | 0.034 | 0.034 |
| csuite_symprod_simpson | 0.318 | 0.695 | 0.487 | 0.427 | 0.819 | 0.160 | 0.487 | 0.487 | 0.819 | 0.168 | 0.101 | 0.242 |
| csuite_weak_arrows | 1.097 | 1.108 | 1.108 | 0.189 | 0.110 | 0.064 | 1.108 | 1.108 | 0.109 | 0.015 | 0.255 | 0.128 |
| csuite_weak_arrows_bt | 0.226 | 0.252 | 0.252 | 0.085 | 0.065 | 0.029 | 0.123 | 0.086 | 0.228 | 0.003 | 0.060 | 0.034 |
| IHDP | 0.187 | 0.187 | 0.187 | 0.090 | 0.101 | 0.116 | 0.187 | 0.187 | 0.187 | 0.187 | 0.146 | 0.087 |
| Twins | 0.030 | 0.025 | 0.025 | 0.022 | 0.025 | 0.025 | 0.068 | 0.025 | 0.022 | 0.042 | 0.022 | 0.060 |

Table 3: Median ATE RMSE data underling our rank table. The median is taken across multiple seeds, with the number of seeds shown in Table 4. Standard deviations are also shown in Table 5. Missing values indicate that the method exceeded the computational budget—this typically occurred for larger graphs.

provides very poor results or fails to find any causal DAGs compatible with the data when working with these datasets. We find both DECI to provide more acceptable performance with the DECI-spline variant producing more reliable results. In this learnt graph setting, causal inference performance deteriorates sharply as a consequence of imperfect causal discovery. However, our findings in terms of relative performance among inference methods stay the same.

### E.3 Discussion of Continuous CSuite Results

1. **lingauss**: When the true graph is available, all our causal inference methods are able to solve this problem. However, when the graph needs to be identified from the data, causal discovery accuracy is around 50%. DECI discovery converges to a posterior with half of its mass on the right distribution resulting in DECI inference methods showing the lowest error.

2. **linexp**: The non-Gaussian noise causes difficulties for DECI-Gaussian, which identifies the wrong orientation in a majority of cases. As a result, inference algorithm yield poor results. Surprisingly, the PC algorithm is also unable to identify the causal graph, leading to overall poor inference performance. With the spline noise model, DECI successfully identifies the causal graph, allowing for all inference algorithms to solve the problem.

3. **nonlingauss**: The non-linear relationship between variables leads all DECI discovery runs to successfully recover the edge direction for this dataset while PC consistently identifies the wrong edge direction. As expected, linear ATE estimation performs poorly on this task. However, we find

| Method / Dataset | DGa | DGa+L | DGa+N | DSp | DSp+L | DSp+N | PC+L | PC+N | T+L | T+N | T+DGa | T+DSp |
|---|---|---|---|---|---|---|---|---|---|---|---|---|
| ER(16, 16) - G | 5 | 5 | 5 | 5 | 5 | 5 | 5 | 5 | 5 | 5 | 5 | 5 |
| ER(16, 16) - S | 5 | 5 | 5 | 5 | 5 | 5 | 5 | 5 | 5 | 5 | 5 | 5 |
| ER(16, 64) - G | 5 | 5 | 5 | 5 | 5 | 5 | 5 | 5 | 5 | 5 | 5 | 5 |
| ER(16, 64) - S | 5 | 5 | 4 | 5 | 3 | 5 | 5 | 5 | 5 | 5 | 5 | 5 |
| ER(64, 64) - G | 5 | 1 | 1 | 5 | 1 | 1 | 4 | 5 | 5 | 5 | 5 | 5 |
| ER(64, 64) - S | 5 | 1 | 1 | 5 | 2 | 2 | 1 | 5 | 5 | 5 | 5 | 5 |
| SF(16, 16) - G | 5 | 2 | 3 | 5 | 4 | 5 | 4 | 5 | 5 | 5 | 5 | 5 |
| SF(16, 16) - S | 5 | 1 | 2 | 5 | 2 | 1 | 3 | 5 | 5 | 5 | 5 | 5 |
| SF(16, 64) - G | 5 | 0 | 0 | 5 | 0 | 1 | 4 | 5 | 5 | 5 | 5 | 5 |
| SF(16, 64) - S | 5 | 0 | 0 | 5 | 0 | 0 | 4 | 5 | 5 | 5 | 5 | 5 |
| SF(64, 64) - G | 5 | 4 | 3 | 5 | 3 | 2 | 3 | 4 | 5 | 0 | 5 | 5 |
| SF(64, 64) - S | 5 | 5 | 5 | 5 | 5 | 5 | 3 | 4 | 5 | 0 | 5 | 5 |
| csuite_cat_to_cts | 20 | 20 | 20 | 20 | 20 | 20 | 20 | 20 | 20 | 20 | 20 | 20 |
| csuite_cts_to_cat | 20 | 20 | 20 | 20 | 20 | 20 | 20 | 20 | 20 | 20 | 20 | 20 |
| csuite_large_backdoor | 20 | 20 | 20 | 20 | 20 | 20 | 20 | 20 | 20 | 20 | 20 | 20 |
| csuite_large_backdoor_bt | 20 | 20 | 20 | 20 | 20 | 20 | 20 | 20 | 20 | 20 | 20 | 20 |
| csuite_linexp | 20 | 20 | 20 | 20 | 20 | 20 | 20 | 20 | 20 | 20 | 20 | 20 |
| csuite_lingauss | 20 | 20 | 20 | 20 | 20 | 20 | 20 | 20 | 20 | 20 | 20 | 20 |
| csuite_mixed_confounding | 20 | 20 | 20 | 20 | 20 | 20 | 20 | 20 | 20 | 20 | 20 | 20 |
| csuite_mixed_simpson | 20 | 20 | 20 | 20 | 20 | 20 | 20 | 20 | 20 | 20 | 20 | 20 |
| csuite_nonlin_simpson | 20 | 20 | 20 | 20 | 20 | 20 | 20 | 20 | 20 | 20 | 20 | 20 |
| csuite_nonlingauss | 20 | 20 | 20 | 20 | 20 | 20 | 20 | 20 | 20 | 20 | 20 | 20 |
| csuite_symprod_simpson | 20 | 20 | 20 | 20 | 20 | 20 | 20 | 20 | 20 | 20 | 20 | 20 |
| csuite_weak_arrows | 20 | 20 | 20 | 20 | 20 | 20 | 20 | 20 | 20 | 20 | 20 | 20 |
| csuite_weak_arrows_bt | 20 | 20 | 20 | 20 | 20 | 20 | 20 | 20 | 20 | 20 | 20 | 20 |
| IHDP | 5 | 5 | 5 | 5 | 5 | 5 | 5 | 5 | 5 | 5 | 5 | 5 |
| Twins | 5 | 5 | 5 | 5 | 5 | 5 | 5 | 5 | 5 | 5 | 5 | 5 |

Table 4: Number of seeds run when computing values in Table 3. For ER/SF graphs, where fewer than 5 seeds were used, this indicates that some runs exceeded the computational budget.

| Method / Dataset | DGa | DGa+L | DGa+N | DSp | DSp+L | DSp+N | PC+L | PC+N | T+L | T+N | T+DGa | T+DSp |
|---|---|---|---|---|---|---|---|---|---|---|---|---|
| csuite_cat_to_cts | 0.107 | 0.210 | 0.108 | 0.105 | 0.194 | 0.172 | 0.000 | 0.000 | 0.011 | 0.011 | 0.000 | 0.000 |
| csuite_cts_to_cat | 0.007 | 0.000 | 0.000 | 0.007 | 0.000 | 0.000 | 0.000 | 0.000 | 0.009 | 0.009 | 0.000 | 0.000 |
| csuite_large_backdoor | 0.724 | 0.737 | 0.724 | 0.041 | 0.062 | 0.022 | 0.000 | 0.000 | 0.046 | 0.028 | 0.055 | 0.031 |
| csuite_large_backdoor_bt | 0.134 | 0.125 | 0.124 | 0.081 | 0.077 | 0.056 | 0.033 | 0.006 | 0.011 | 0.010 | 0.037 | 0.004 |
| csuite_linexp | 0.013 | 0.000 | 0.000 | 0.014 | 0.000 | 0.000 | 0.000 | 0.000 | 0.015 | 0.015 | 0.000 | 0.000 |
| csuite_lingauss | 0.435 | 0.498 | 0.432 | 0.454 | 0.355 | 0.489 | 0.000 | 0.001 | 0.020 | 0.022 | 0.000 | 0.000 |
| csuite_mixed_confounding | 0.371 | 0.389 | 0.353 | 0.369 | 0.418 | 0.422 | 0.000 | 0.000 | 0.010 | 0.010 | 0.000 | 0.000 |
| csuite_mixed_simpson | 0.524 | 0.522 | 0.507 | 0.585 | 0.522 | 0.514 | 0.000 | 0.000 | 0.010 | 0.008 | 0.000 | 0.000 |
| csuite_nonlin_simpson | 1.071 | 0.593 | 0.976 | 1.080 | 1.326 | 1.409 | 0.000 | 0.000 | 0.283 | 0.304 | 0.000 | 0.000 |
| csuite_nonlingauss | 0.014 | 0.000 | 0.000 | 0.016 | 0.000 | 0.000 | 0.000 | 0.000 | 0.013 | 0.016 | 0.000 | 0.000 |
| csuite_symprod_simpson | 0.187 | 0.162 | 0.162 | 0.180 | 0.000 | 0.128 | 0.000 | 0.000 | 0.090 | 0.124 | 0.000 | 0.000 |
| csuite_weak_arrows | 0.433 | 0.401 | 0.426 | 0.256 | 0.118 | 0.075 | 0.000 | 0.000 | 0.160 | 0.131 | 0.000 | 0.008 |
| csuite_weak_arrows_bt | 0.104 | 0.118 | 0.113 | 0.093 | 0.067 | 0.067 | 0.000 | 0.002 | 0.020 | 0.017 | 0.000 | 0.001 |
| IHDP | 0.021 | 0.000 | 0.062 | 0.013 | 0.048 | 0.037 | 0.024 | 0.024 | 0.014 | 0.022 | 0.000 | 0.000 |
| Twins | 0.018 | 0.000 | 0.000 | 0.004 | 0.022 | 0.019 | 0.023 | 0.024 | 0.003 | 0.009 | 0.000 | 0.000 |

Table 5: Standard deviations for ATE RMSE results.

DoWhy non-linear to not fare much better, likely this is because DML still assumes a linear relationship between treatment and target. DECI solves the task successfully.

Table 6: Method rank on different (CSuite, Twins, IHDP) datasets, ranking by median ATE RMSE. We present the median of the rank over 15 datasets. Supporting data in Table 3. We treat methods with access to the true graph separately. Note that methods without access to the true graph would typically have a minimum possible rank of 5.

| Method | Mean rank |
|---|---|
| DECI Gaussian (DGa) | 8 |
| DECI Gaussian DoWhy Linear (DGa+L) | 9.5 |
| DECI Gaussian DoWhy Nonlinear (DGa+N) | 9.5 |
| DECI Spline (DSp) | 6 |
| DECI Spline DoWhy Linear (DSp+L) | 6 |
| DECI Spline DoWhy Nonlinear (DSp+N) | 4.5 |
| PC + DoWhy Linear (PC+L) | 9 |
| PC + DoWhy Nonlinear (PC+N) | 9 |
| True graph DECI Gaussian (T+DGa) | 2 |
| True graph DECI Spline (T+DSp) | 3 |
| True graph DoWhy Linear (T+L) | 4 |
| True graph DoWhy Nonlinear (T+N) | 4 |

4. **nonlin_simpson**: Even when the true graph is available, none of our inference methods are able to recover the true ATE on this more difficult task. We observe non-linear methods (DECI and DoWhy-nonlinear) to perform similarly to each other and more strongly than the simple linear adjustment. For the CATE task, the true value is close to 0. This is correctly identified by both DECI-Gaussian and DECI-Spline. Interestingly, we find both linear and non-linear DoWhy variants to overestimate the causal effect when using the backdoor criterion. We attribute this to DECI solving a lower dimensional problem when estimating CATE. While DECI simply regresses the conditioning variable onto the effect variable. The backdoor adjustment employed by DoWhy requires regression from the joint space of conditioning variables and confounders onto the effect variables. The latter procedure involves estimating the relative strength of confounders and conditioning variables, which is a more challenging task.

    This dataset provides a challenging causal discovery task. DECI identifies the correct edges with probability 0.9. It capacity to recover the edge orientation is slightly worse 0.65. This imperfect causal discovery leads to poor inference for all methods. (potentially because they get Simpson's paradox the wrong way around).

5. **symprod_simpson**: Even with access to the true graph, no inference method is able to solve this problem. However, we find non-linear methods to clearly outperform linear adjustment for both CATE and ATE estimation. Among non-linear methods, performance is similar for ATE estimation, with DECI-Gaussian performing slightly better than DoWhy-nonlinear and DECI-Spline slightly worse. However, when estimating CATE, DECI inference present an error twice as low as nonlinear DoWhy. Again, we attribute this to the backdoor adjustment employed by DoWhy being a more challenging inference task than the 1d regression on simulated data employed by DECI.

    In terms of causal discovery, results are similar to nonlin-simpson with PC failing completely and DECI obtaining an adjacency score of 0.92 and orientation of 0.7. The imperfect graph knowledge hurts causal inference. Again we see the non-linear backdoor adjustment to perform similarly to DECI for ATE estimation while DECI shows decisively stronger performance when estimating CATE. As expected, the linear adjustment method fares poorly in this strongly non-linear setting.

6. **weak_arrows**: When the true causal DAG is available we find that both DoWhy methods solve this ATE problem while both DECI methods predict slightly suboptimal ATE values.

    In terms of causal discovery, DECI clearly outperforms PC with the spline noise model again proving more reliable and leading to better ATE estimates. Although no methods are able to solve the task, we find that non-linear DoWhy with the DECI-spline graphs performs best. We hypothesize that

the amortised function structure employed by DECI suffers in very densely connected graphs with weak edges, like is the case here.

7. **large_backdoor**: With access to the true graph, DECI methods outperform both Dowhy variants for both ATE and CATE estimation. DECI-spline performs best and is able to solve both problems. When faced with many confounders, adjustment procedures suffer from large variance. As a result, despite the non-linearity of the functional relationships at play, the simpler linear backdoor adjustment outperforms the non-linear DML approach. On the other hand, DECI's simulation based approach is not disadvantaged in this setting.

Following the trend of the previous datasets, PC performs poorly in terms of causal discovery, biasing downstream inference methods which perform poorly in terms of ATE and CATE estimation. DECI discovery is more reliable, an effect most noticeable when using the spline noise models. With the DECI-Spline posterior over graphs, both DECI-spline and linear DoWhy are able to solve the ATE problem and DECI-spline is the only method capable of solving the CATE task. For both tasks and noise models DECI outperforms non-linear DoWhy, again showing its invariance to the size of potential adjustment set.

### E.4 Discussion on discrete variables

1. **cat_to_cts** We can see when the graph is available, $T+DGa$ performs the best indicating the flexibility of DECI's SEM and effectiveness of the training algorithm. However, when the graph is not available, the performance drops for all methods. This indicates that the discovery part of DECI indeed suffers from such model mis-specification. Changing to discrete variables breaks the model identifiability, which may result in multiple graphs to be learned. This can be partially verified in Table 5 with high variance for DECI methods.

2. **cts_to_cat** The performances drop is not as severe as **cat_to_cts**. From Table 5, the variance is much smaller. This indicates that although this breaks the identifiability, DECI still tends to recover a unique graph. As for the detailed analysis on this scenario, we can leave that for future work.

3. **large_backdoor_bt** The performance drop is also minor. From Table 5, we can see that the variance is moderate compared to the identifiable case **large_backdoor**, indicating that DECI still tends to find a unique graph. This shows certain robustness of DECI to model mis-specification.

4. **mixed_simpson** The performance significantly drops compared to the one with **True graph**. The high variance in Table 5 also suggests that multiple graphs may be found due to model unidentifiability.

5. **weak_arrows_binary_t** The performance drop is moderate and also the variance is not high, indicating certain robustness to this mis-specification.

6. **mixed_confounding** The performance drops and the variance is high, showing that DECI cannot recover the unique graph and is sensitive to this type of mis-specification.

From the above analysis of data type mis-specification, we can see DECI shows certain robustness. This opens the door for future directions regarding which type of data mis-specification DECI is sensitive to and why.

### E.5 Synthetic Graph Experiments

We test the performance of DECI on ATE estimation with random graphs as described in section 4.1. For each graph, we randomly generate interventional data for up to five random interventions. We chose the effect variable as the last variable in the causal order that has not yet been used for data generation. For each effect variable we chose the intervention by randomly traversing the graph up to three edges away from the effect variable.

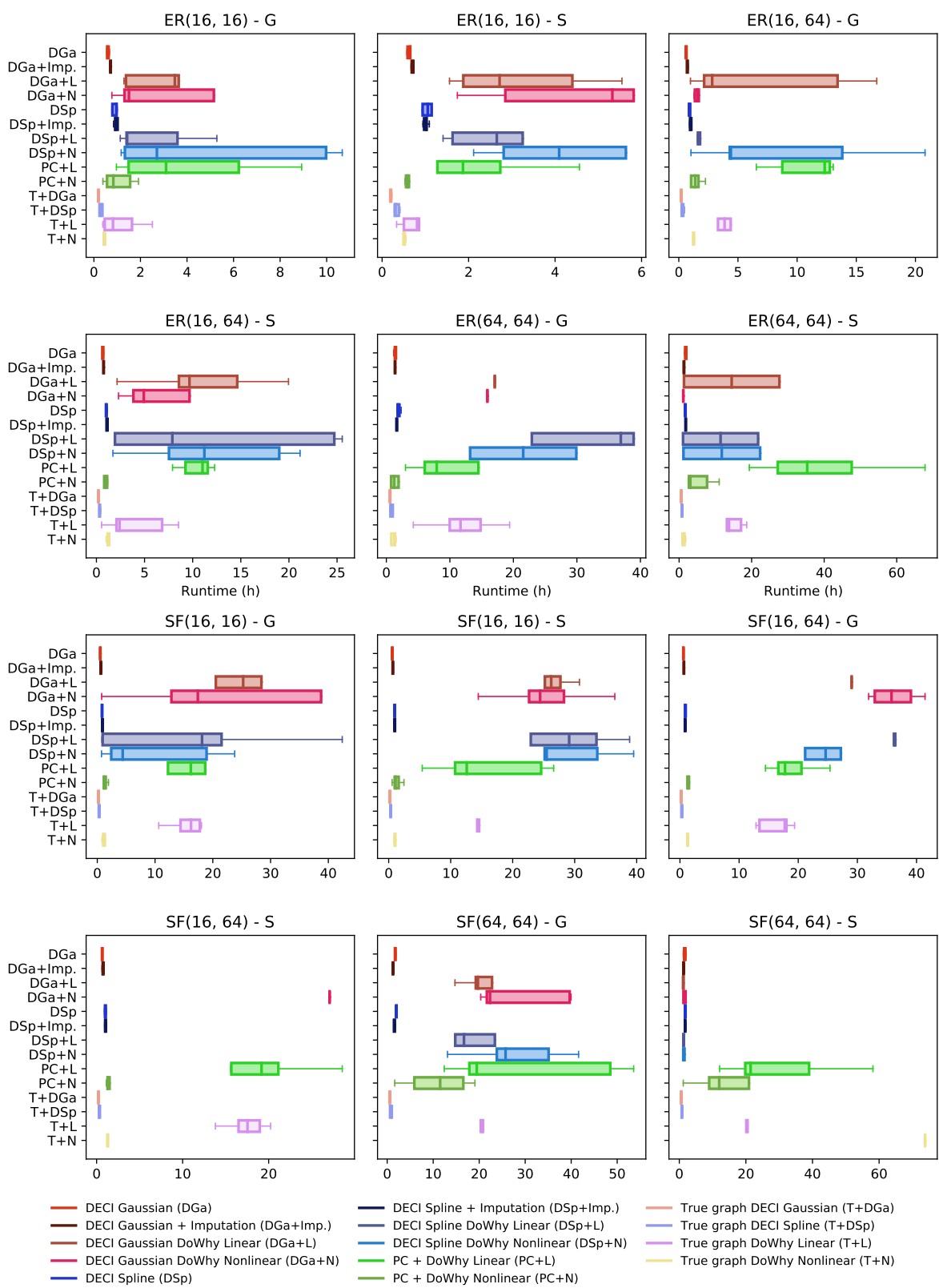

Figure 11: Runtime of End-to-end ATE estimation methods on synthetic graphs.

Table 3 shows the performance of the ATE estimation of DECI and all baselines on the synthetic graph data. We only show results for methods that have a runtime of less than one day. Figure 11 shows the runtimes for the different methods. DECI has consistently the lowest runtime and scales best to larger graphs. While the runtime of DECI stays approximately constant for various graphs, the runtime of the ATE estimation baselines increases with more complex graphs. In general, the methods using the true graph outperform the methods that also perform causal discovery. Further, no method strongly outperforms all other methods with DECI being a strong competitor to the already established DML methods. Lastly, we can see that DECI is capable of performing causal discovery, data imputation and ATE estimation in an end-to-end fashion without degrading performance.

### E.6 Learning in Non-identifiable Settings with the Help of Graph Priors

We investigate the utility of prior knowledge over causal graphs for causal discovery and end2end inference in non-identifiable and difficult to identify settings. Specifically, we generate 2 datasets composed of 2000 training examples each. The first is composed of only linear relationships between variables and Gaussian additive noise, making the causal graph non-identifiable. The second dataset also uses linear functions but has a mix of exponential and Tanh-Gaussian noise. Although identifiable, discovery in this latter setting is challenging.

We introduce prior knowledge about graph sparseness through the weighted adjacency matrix $W_0 \in [0,1]^{D \times D}$, with zero entries encouraging sparser graphs. The resulting informed DECI prior is

$$p(G) \propto \exp\left(-\lambda_s \|G - W_0\|_F^2 - \rho\, h(G)^2 - \alpha\, h(G)\right),$$

with the scalar $\lambda_s$ regulating the strength of the prior beliefs encoded in $W_0$.

We compare DECI inference with access to the true graph to end2end DECI inference. In the latter case we consider a PC prior, which has as its mean the CP-DAG provided by PC. We consider different prior strengths, i.e. the value of the entries of $W_0$, between 0 and 1. We also experiment with introducing the true-graph as a prior of this form, yielding what we refer to as the "informed prior".

In the non-identifiable case, we find both DECI (prior strength 0) and PC discovery to provide incorrect graphs. Interestingly, providing the PC CPDAG as a prior for DECI can yield large gains in terms of causal discovery due to a variance reduction effect. These gains do not translate to better ATE estimation, where performance is not improved over the uninformative prior. Providing knowledge of the true graph does help causal inference, with a more confident prior yielding better results.

In the difficult identifiable case, the PC prior does provide gains to DECI. We find the optimal prior strength to be 0.5: a balanced combination of PC and DECI discovery is most reliable, while using exclusively one of the two algorithms yields worse results. In this identifiable setting informed DECI discovery is able to obtain perfect ATE estimation performance with a prior strength as low as 0.2.

### E.7 Understanding Robustness to Unobserved Confounders

To evaluate the robustness of DECI to the presence of unobserved confounders, we introduced a *new* CSuite datasets. **Fork nonlin. Gauss** has the graph shown in Figure 14a and uses nonlinear functions with Gaussian exogenous noise. We trained DECI on the data from observed variables only (as shown in Figure 14), as well as on the full dataset treating all variables as observed. Table 7 shows the results.

Table 7: ATE RMSE results on CSuite Fork nonlin. Gauss dataset with unobserved confounders.

| Method | ATE RMSE (mean ± s.e. from 5 seeds) |
|---|---|
| DECI Gaussian (DGa) | 0.034 ± 0.007 |
| DECI Spline (DSp) | 0.036 ± 0.009 |
| DECI Gaussian (DGa) full observability | 0.019 ± 0.002 |
| DECI Spline (DSp) full observability | 0.018 ± 0.002 |

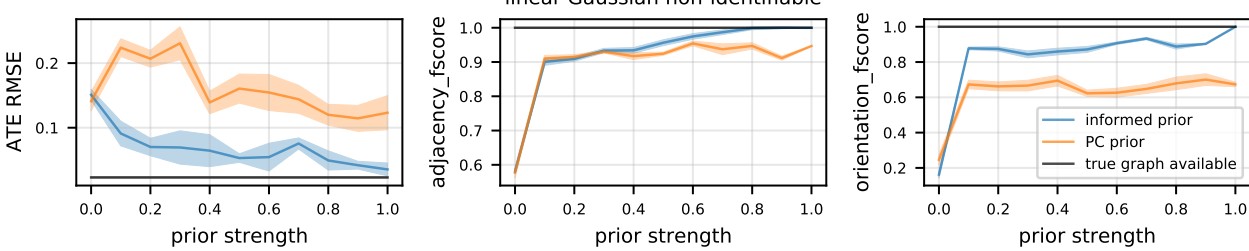

Figure 12: Causal discovery and inference results obtained on a 9 node linear Gaussian dataset, where the graph is non-identifiable without prior knowledge. We perform DECI inference with the true graph and DECI end2end inference with different priors. Informed prior refers to using a smoothed version of the true graph as the prior. PC prior refers to using the CP-DAG outputted by PC as the prior mean $W_0$. The prior strength indicates how much prior mass is placed on the mean prior graph $W_0$ and how much is spread across all other DAGs.

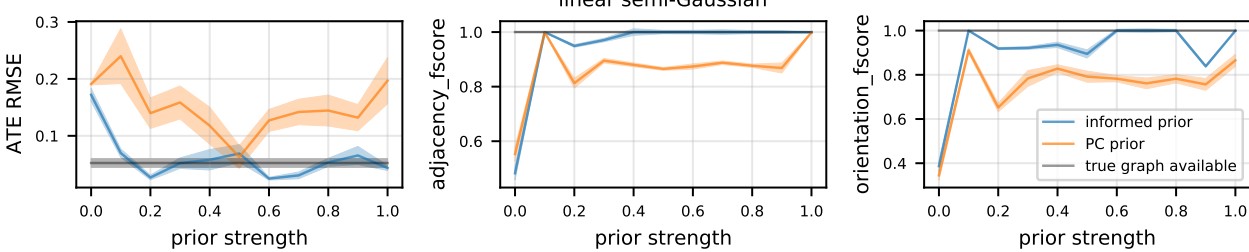

Figure 13: Causal discovery and inference results obtained on a 9 node linear non-Gaussian dataset, difficult to identify without prior knowledge. We perform DECI inference with the true graph and DECI end2end inference with different priors. Informed prior refers to using a smoothed version of the true graph as the prior. PC prior refers to using the CP-DAG outputted by PC as the prior mean $W_0$. The prior strength indicates how much prior mass is placed on the mean prior graph $W_0$ and how much is spread across all other DAGs.

## F Additional Related Work

Similarly, recent works have combined the continuous DAG learning approach with variational inference (Lorch et al., 2021; Annadani et al., 2021; Cundy et al., 2021; Deleu et al., 2022). However, most of them focus on linear Gaussian models (Annadani et al., 2021; Cundy et al., 2021; Deleu et al., 2022). Only Lorch et al. (2021) considers non-linear functional forms but is constrained to Gaussian noise models. To the best of our knowledge, we are the first to present an identifiability proof for causal discovery using VI.

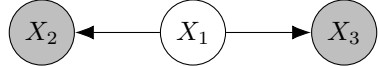

(a) Fork. Treatment $X_2$, outcome $X_3$.

Figure 14: CSuite graphs with unobserved confounders.

