# OpenReview forum: "Deep End-to-end Causal Inference"
_TMLR — Accepted by TMLR_

### Review · Reviewer_DoeS · 2024-03-02

**Summary Of Contributions:**

The paper introduces the Deep End-to-end Causal Inference (DECI) framework, proposed to address challenges in causal inference. The proposed framework is a flow-based non-linear additive noise model combined with variational inference, enabling Bayesian causal discovery and inference. The framework unifies many existing structural equation model (SEM) based causal inference techniques and can recover the ground truth mechanism under standard assumptions. Additionally, DECI can be extended to handle heterogeneous, mixed-type data with missing values, allowing for both continuous and discrete treatment decisions. Extensive empirical evaluations have been conducted to demonstrate DECI's competitive performance compared to relevant baselines for both causal discovery and inference.

**Audience:**

Yes

**Broader Impact Concerns:**

No concerns about broader impacts.

**Claims And Evidence:**

Yes

**Requested Changes:**

See weaknesses.

**Strengths And Weaknesses:**

I have read some previous versions of this paper and seen the improvements made by the authors. This paper contributes to integrating the causal structure learning task and the downstream causal inference task into a unified learning framework, which is quite meaningful from my perspective. The paper is very well written and polished. The topic is of theoretical and practical importance. Moreover, the authors have provided really extensive experimental results to show the effectiveness of the proposed method, making this paper quite solid.

However, there are a few weaknesses in the paper that could be addressed:

1. The authors should clarify why using the learned graph makes treatment effect estimation more efficient. While prior knowledge of treatment and outcome can be used to estimate Conditional Average Treatment Effects (ATE) and ATE directly using methods like Slearner, Tlearner, and Xlearner, it would be beneficial to explain how the learned graph improves efficiency in this context.

2. Equation 9 restricts the noise distributions to be a fixed Gaussian, which may not accurately reflect real-world scenarios where noise distributions are not equal-variance Gaussian. It would be helpful if the paper discussed how this restriction affects the learning results of the proposed method.

3. The method's effectiveness in causal discovery may be limited when the dimensionality of the variables is high (more than 100), which could impact the results of causal inference. It would be informative to see the results of this scenario and compare the method with well-known Targeted Maximum Likelihood Estimation (TMLE) methods, such as TarNet.

4. In Figure 3, the probabilistic property allows the method to incorporate the results of multiple learned causal graphs to estimate treatment effects. However, if one of the learned graphs is incorrect but has a lower probability, it would be helpful to discuss how this would affect the inference results.

Additionally, it would be beneficial to elaborate more on the similarities between the proposed method and "Deep Structural Causal Models for Tractable Counterfactual Inference" to provide a clearer understanding of the contributions of the proposed method to existing literature.

---

> ### Author Response · Authors · 2024-04-11
> **Authors' response 1**
>
> Thanks for reviewer’s suggestions. We will try to address your concerns in the following.
>
> 1. > The authors should clarify why using the learned graph makes treatment effect estimation more efficient ...
>
> The main reason on why causal graph can help causal inference is the following:
>
> For causal inference approach, like TLearner, SLearner, etc., also assume an implicit causal graph. For example, covariates act as confounders, and there is a direct connection from the treatment to the effect variable. However, how to pick the covariate variable is challenging, and most approaches simply pick all other variables, leading to an incorrect graph in most cases. In addition, strong prior knowledge is also required to propose a reasonable graph. For example, it is hard in general to find all possible mediators between treatment and effect. However, DECI relieves this problem by incorporating a graph learning stage. DECI can leverage the identified graph distribution to estimate the effect to target variables. Namely, it can pick a reasonable set of variables as the covariates based on the graph. Also, DECI can naturally deal with mediators.
>
> 2. > Equation 9 restricts the noise distributions to be a fixed Gaussian ...
>
> We want to clarify that we do not assume a fixed Gaussian noise for the DECI formulation. We specifically mentioned this in our contribution 1: “we extend the noise distribution beyond standard Gaussian and propose a flexible noise based on spline transformation.". In equation 9, we also have the Jacobian determinant because we assume the noise is an invertible transformation of the Gaussian (e.g. spline flow in DECI). We assume a Gaussian distribution as the base distribution for simplicity, but one can easily replace it with another form of distribution as long as there exists a differentiable density. We have elaborated on this in the revised paper.
>
> 3. > The method's effectiveness in causal discovery may be limited when the dimensionality of the variables is high (more than 100) ...
>
> Thanks for the suggestion with high dimensional data. We want to emphasise that one of the main advantages of DECI is to provide graphs, which is an important causal quantity to estimate. However, the inference approaches, like TarNet, lack the ability to do this. Apart from our end-to-end pipeline, one can also concatenate a discovery approach (like PC) with an inference approach, like what we have reported in the experiment. For lower dimensional settings, our experiments has demonstrated better performance. Since the accurate graph learning is a pre-requisite for the inference stage, we have conducted additional experiment on the graph discovery with $100$ dimensional data to demonstrate the relative robustness compared to the baselines.
>
> We compared DECI with PC algorithm on $100$ dimensional data. We use Erdos-Reyni graph distribution to randomly sample a graph with $100$ nodes. We use random Fourier features to generate the functional relationship between nodes and add Gaussian noise. The total number of edges is $287$. For DECI, we use the Gaussian noise distribution with learnable variance, and neural network for functional relationships. For PC, we use Fisher-z test with significance level $0.05$.
>
> |      | Adjacency F1 | Orientation F1 |
> |------|--------------|----------------|
> | PC   | 0.398        | 0.203          |
> | DECI | 0.638        | 0.536          |
>
> We can observe that DECI significantly outperforms PC, which demonstrates certain robustness in high dimensional settings.
>
> 4. > In Figure 3, the probabilistic property allows the method to incorporate the results of multiple ...
>
> Thanks for the reviewer’s suggestion. The way that DECI estimates causal quantities is based on the Monte Carlo (MC) approximation over the graph distribution. Therefore, the quality of the inference depends heavily on the graph distribution and the number of samples used during MC. If the incorrect graph has very low probability such that it is rarely sampled, then the estimation quality will not be affected with smaller graph sample size. But the variance of the quantity will be high. If the graph sample size increases towards infinity, the estimation variance will tend towards zero, but the incorrect graph will have innegligible effect. We have added this to the revised paper.

---

> > ### Comment · Reviewer_DoeS · 2024-04-11
> > **Thanks**
> >
> > Thank you for your response, which has addressed my concerns. Happy to see that this work is ready for publication.

---

> ### Author Response · Authors · 2024-04-11
> **Authors' response 2**
>
> 5. > it would be beneficial to elaborate more on the similarities between ....
>
> The work “Deep Structural Causal Models for Tractable Counterfactual Inference” is orthogonal to DECI. We will call it DSCM in the following. The main difference is that DSCM tries to solve the noise inference problem during the counterfactual estimation. In particular, they aim to infer the posterior of the noise given observation using variational inference, and then apply it in the Abduction and Prediction steps. They assume that the causal graph is known. On the other hand, DECI can infer the causal graph from the observational data and also estimate the counterfactual quantity (we do not report in the paper, but this is a straight-forward extension). Due to the additive noise nature of DECI, the posterior noise is analytically tractable and does not need the DSCM inference technique. Although one can easily combine DECI with DSCM to extend beyond additive noise SCM, we will lose the graph identifiability property, and there will be no guarantee on the graph correctness. We have added this discussion in the related work.

---

### Review · Reviewer_Xkxj · 2024-03-10

**Summary Of Contributions:**

This study introduces a deep-learning-based framework that spanning from causal discovery to causal effect estimation. Operating under specific structural assumptions, including the absence of unmeasured confounders, and making parametric assumptions about noise, it employs the NOTEARS-based approach for causal discovery. Following this, the framework uses the posterior distribution over the graph, denoted as q_{\phi}(G), to derive the interventional Structural Causal Model (SCM). This facilitates the estimation of causal effects from these models. The authors highlight the practical advantages of their proposed model through a variety of examples, underlining its applicability and effectiveness.

**Audience:**

Yes

**Broader Impact Concerns:**

This work doesn't possess any ethical concerns.

**Claims And Evidence:**

No

**Requested Changes:**

Please address the concerns described in the Weakness and Minor issues.

**Strengths And Weaknesses:**

# Strength
The work effectively demonstrates the practical utility of its framework through a range of examples, from synthetic to real-world scenarios. Its most notable strength lies in providing a comprehensive end-to-end solution for causal inference, encompassing both causal discovery and estimation processes.


# Weakness
The overall soundness of the proposed framework is questionable, primarily due to its reliance on a series of approximations and assumptions. This raises doubts about the reliability of the resulting estimated Average Treatment Effect (ATE) / Conditional Average Treatment Effect (CATE). Compared to Markov-equivalence-class (MEC)-based methods like those of Jung et al., 2021, and Guo et al., 2021, this framework is predicated on more stringent assumptions, such as the absence of unmeasured confounders, third-order differentiability of functions, specific noise structures, and correctly specified models. In contrast, the existing methods utilizing the Partial Ancestral Graph (PAG) or Completed Partially Directed Acyclic Graph (CPDAG) only rely on the assumption of a correct Conditional Independence (CI)-oracle.

Furthermore, the reliance on the NOTEARS method, an approximate approach for discovering the true causal graph, adds another layer of uncertainty. The method is possible to select a graph outside the Markov equivalence class, which may lead to a misspecified graph. Given that this approximation influences both the learning of the graph and the estimation of causal effects, the resultant causal effect estimate is not reliable.

Finally, the paper falls short in adequately comparing its approach with previous works. For instance, Jung et al., (2021b) and Guo et al., (2021) also offer end-to-end causal inference frameworks, leveraging the PAG, a summary of the MEC allowing for unmeasured confounders. These methods, utilizing readily available causal discovery techniques, warrant a more transparent acknowledgment and detailed comparative analysis.

# Minor issues

1. The articulation of the contributions could be more accessible to readers. The paper should specifically elaborate on the nature of the graph posterior in the contribution section.
2. The absence of theoretical guarantees regarding the convergence of the estimated causal effect to the true causal effect, including the convergence rate, is a significant omission.
3. Clarity is needed on what novel contributions this method provides, especially in comparison to existing methods like those of Jung et al., (2021b) and Guo et al., (2021). What distinguishes this method from others in the field?

---

> ### Author Response · Authors · 2024-04-11
> **Authors' response 1**
>
> Thanks for reviewer's comments. We will try to answer your concerns in the following.
>
> 1. > The overall soundness of the proposed framework is questionable, primarily due to its reliance on a series of approximations and assumptions. ...
>
> Regarding the assumptions, we would like to clarify the following:
> - Our main assumptions are commonly used canonical assumptions in the functional causal discovery literature and inherit from the well-known work of [1].
> - Our assumptions are not directly comparable to those of Jung et al., 2021, and Guo et al., 2021, since our result provides stronger statements. In fact, our result ensures the unique identifiability of all quantities, including causal graph $G$, as well as causal effects. While Jung et al., 2021, and Guo et al., 2021 seem to impose less assumptions, they do require access to a given PAG to start with (which is hard to obtain in practice), and only work with set-identifiability. In fact, Guo et al., 2021 enumerates all possible effects, and Jung et al., 2021 only returns effect estimates if it is identifiable given the PAG. Our assumptions will be significantly weaker if we relax the requirement to the same level of set-identifiability.
> - Moreover, the above mentioned work also requires assumptions that are in the similar level as us . For example, Guo et al., 2021 in fact assumes no latent confounding; Jung et al., 2021 assumes all variables to be discrete. In fact, our framework can be generalized to handle latent confounders, but these are beyond the scope of a single paper.
>
> 2. > Furthermore, the reliance on the NOTEARS method, an approximate approach for discovering the true causal graph, adds another layer of uncertainty. ...
>
> We want to clarify the following regarding this concern:
>
> - We admit that the misspecified causal graph from NOTEARS is possible in practice. For example, limit training data, misspecified model assumption and sub-optimal solutions can all lead to incorrect graphs to be discovered. However, we think this should not be regarded as the limitation of DECI. In fact, all causal discovery methods rely on certain assumptions that may not be met in practice. For instance, the conditional-independence based methods require the effectiveness of CI-test. For example, [2] proposes a kernel-based independence test, where its asymptotic distribution only holds true when the number of test samples $\rightarrow \infty$. To the best of my knowledge, all conditional independence tests in causal discovery rely on assumptions about asymptotic statistics or kernel space, requiring infinite samples or characteristic kernels.
>
> - To improve the reliability of DECI due to the finite training data, we propose to adopt the Bayesian setup, where a posterior distribution of the graph is learned instead of a single graph (compared to NOTEARS), so that DECI can reason about causal quantities with uncertainty.
>
> - Since all methods require assumptions that are not met in practice, we can only measure the effectiveness through extensive experiments. This is exactly one of our contributions that has been recognised by other reviewers. Specifically, we conducted comprehensive experiments to demonstrate the effectiveness of DECI (refer to section 4 and appendix E). In particular, we presented many combinations of causal discovery and inference approaches, including the MEC-based discovery (e.g. PC) and Double machine learning. From Table 1 and Table 6 (Appendix E.2), we can see that DECI-based approach achieves higher performance rank compared to PC+DML, suggesting the DECI's reliability in practice.
>
> - Apart from the demonstrated practical advantage, let's consider the ideal case where all assumptions are met (assumptions that are very common in SEM-based causal discovery [1]). Theorem 1 of DECI provides a guarantee of recovering the true graph. Compared to MEC, DECI provides stronger results regarding the identifiability of causal quantities as previously discussed.
>
> Reference:
>
> [1] Peters, Jonas, et al. "Causal discovery with continuous additive noise models." (2014).
>
> [2] Zhang, Kun, et al. "Kernel-based conditional independence test and application in causal discovery." arXiv preprint arXiv:1202.3775 (2012).

---

> ### Author Response · Authors · 2024-04-11
> **Authors' response 2**
>
> 3. > Finally, the paper falls short in adequately comparing its approach with previous works ...
>
> We would like to emphasise that both Jung et al., (2021b) and Guo et al., (2021) are not applicable in our setting.
> - Jung et al., (2021b) assumes discrete variables and does not apply to our continuous setting.
> - Jung et al., (2021b) has much higher computational complexity ($\mathcal{O}(D^4)$ where $D$ is the number of variables). It has been only tested on datasets of variable size less than 7. Therefore, there was very little empirical evidence to justify the claim that it "offers end-to-end causal inference frameworks".
> - Guo et al., (2021) in fact does **NOT** support unmeasured confounders, as opposed to the reviewer's comment. If we do assume no unmeasured confounders, Guo et al., (2021) is orthogonal to our work, as it mainly addresses the issue of minimal enumeration problem of possible ATEs. Since minimal enumeration is not our main concern, our baseline (PC + DML or PC + DECI) already addresses similar settings, where we enumerates possible DAGs in a discovered MEC, and return the average total effect.
>
> 4. > (1) The absence of theoretical guarantees regarding the convergence ...; (2) Clarity is needed on what novel contributions this method provides ...
>
> Thanks for the reviewer's suggestion, we will address in the following:
> 1. It is true that Theorem 1 only guarantees the recovery of the ground truth graph at the global optimum, rather than focusing on the convergence rate. Showing the convergence rate itself can be a separate research topic due to the technical difficulty. Our approach adopts a variational formulation, putting another layer of difficulty. There is existing analysis regarding the convergence of VI [3], but they are not directly applicable to discrete objects like graphs. Thus, showing the convergence rate of DECI requires new tools to be designed for discrete objects and is of an independent interest for VI that is out of the scope of this project.
>
> 2. As previously discussed, our approach cannot be directly compared to Jung 2021b. Specifically, Jung requires a PAG as a starting point, which is typically unavailable in practice. On the other hand, DECI can infer a distribution over DAGs from the observational data. Thus, Jung is a causal inference method, but DECI is a combination of causal discovery and inference together. Jung also requires the causal quantity to be identifiable from PAG as a prerequisite (e.g. ITE can not be estimated by Jung’s approach), but DECI, due to its SEM formulation, can identify all causal quantities automatically.  Last but not least, Jung relies on DML as the estimator, which is linear in nature, whereas DECI adopts a more flexible non-linear neural networks.
> DECI cannot be directly compared to Guo 2021 as well. Firstly, Guo 2021 also assumes an MPDAG as the input, which is not available in general. Guo 2021 also assumes a set identifiability regarding the possible DAG set from MPDAG. DECI, on the other hand, provides a stronger identification to all causal quantities. DECI leverages the neural network to model the functional relationship between variables, whereas Guo 2021 uses non-parametric estimators for causal quantities. We will add those discussions in the revised version of the paper.

---

> ### Comment · Reviewer_Xkxj · 2024-04-14
> **Response**
>
> > Our assumptions are not directly comparable to those of Jung et al., 2021, and Guo et al., 2021, since our result provides stronger statements. In fact, our result ensures the unique identifiability of all quantities, including causal graph, as well as causal effects.
>
> 1. I don't understand what you mean by the unique identifiability (versus set-identifiability). Does it mean that any causal quantity is identifiable? Does the proposed method can handle the case where the causal effect is unidentifiable?
>
> 2. I disagree with this statement that the authors' result provides stronger statements. Specifically, the access to the PAG can be made without loss of generality, because PAGs can be learned from the data.
>
> > "However, we think this should not be regarded as the limitation of DECI."
>
> ---
>
> Here is a follow-up question. Have you considered the case, in which case the graph is unidentifiable? What would the proposed method behave if the causal quantity is not identifiable from the learnt graph?
> 1. I disagree with the statement. In conditional-independence-based methods, the learning guarantees (detailing under which conditions and at what convergence rate the causal effect can be revealed) are well established. For instance, the condition for the kernel-CI is quite transparent, as it requires that the SEM functions belong to specific RKHS spaces. However, I am uncertain whether the NOTEARS method provides such learning guarantees, even when the sample size increases indefinitely. The assumptions in Theorem 1, such as "correctly specified models and no unmeasured confounders," are not equivalent to assumptions found in methods like kernel-CI, where it is stipulated that "f is in Sobolev spaces," because the Theorem 1 assumption is less transparent and overly idealized.

---

> > ### Author Response · Authors · 2024-04-17
> > **Authors' response 1**
> >
> > Thank you for commenting on our rebuttal.
> > > I don't understand what you mean by the unique identifiability (versus set-identifiability). Does it mean that any causal quantity is identifiable? Does the proposed method can handle the case where the causal effect is unidentifiable?
> >
> > - Unique identifiability means that in our results, we aim at ensuring the causal effect and causal graph are identifiable (i.e., uniquely recovered from observational distribution). On contrary, Guo et al., 2021 for example only estimates a set of the non-unique effects compatible with data (i.e., non-unique recovery, or set-idetifiability).
> > -  These work (Jung et al., 2021, Guo et al., 2021) technically does not "handle causal effect that is unidentifiable", but just simply does not return an estimate or enumerating the possible effects. In this sense, our method can still be applied since our Bayesian formulation naturally handles this setting by returning a distribution of non-unique effects or graphs.
> >
> > In fact, Guo et al., 2021 enumerates all possible effects, and Jung et al., 2021 only returns effect estimates if it is identifiable given the PAG. Our assumptions will be significantly weaker if we relax the requirement to the same level of identifiability.
> >
> > > I disagree with this statement that the authors' result provides stronger statements.
> >
> > As explained above, our result aims at stronger identifiability instead of set-identifiability. We acknowledge that the reviewer might not agree that authors' result provides stronger statements. However, we believe it is clear that the assumptions in our work is not strictly stronger than previous work (Jung et al., 2021, Guo et al., 2021) either. For example, as mentioned in our previous rebuttal, Jung et al., 2021 assumes all variables to be discrete; Guo et al., 2021 in fact assumes no latent confounding etc;
> >
> > > Specifically, the access to the PAG can be made without loss of generality. PAGs can be learned from the data.
> >
> > - In fact, Guo et al., 2021 assumes access to MPDAG, not PAG, which relies on non-trivial assumptions such as no latent confounders.
> > - While PAGs in principle can be learned based on conditional independence, making use of them is non trivial: 1, they cannot distinguish between causal graphs that entail the same sets of conditional independence; 2, in practice, learning an accurate PAG from data can be very difficult, and can under-perform the functional model-based methods (Maeda et al, 2020, 2021) and differentiable methods (Bhattacharya et al., 2021), which is the type of methods that we focus in this paper.
> >
> > References:
> >
> > Maeda, Takashi Nicholas, and Shohei Shimizu. "Causal additive models with unobserved variables." Uncertainty in Artificial Intelligence. PMLR, 2021.
> >
> > Maeda, Takashi Nicholas, and Shohei Shimizu. "RCD: Repetitive causal discovery of linear non-Gaussian acyclic models with latent confounders." International Conference on Artificial Intelligence and Statistics. PMLR, 2020.
> >
> > Bhattacharya, Rohit, et al. "Differentiable causal discovery under unmeasured confounding." International Conference on Artificial Intelligence and Statistics. PMLR, 2021.

---

> > ### Author Response · Authors · 2024-04-17
> > **Authors' response 2**
> >
> > > Have you considered the case, in which case the graph is unidentifiable? What would the proposed method behave if the causal quantity is not identifiable from the learnt graph?
> >
> > As we have discussed in the previous response, our Bayesian setup naturally handles this setting by returning a distribution of non-unique effects or graphs. Specifically, we have a dedicated section in our paper (Appendix E.6) to talk about the the unidentifiability of graphs. We generate an unidentifiable dataset with linear Gaussian additive noise model, and conduct studies with/without the help of priors. In a nutshell, if the graph is un-identifiable, this means the underlying data generation mechanism cannot be represented by DECI since DECI is structure identifiable due to its additive noise nature with nonlinear functions. This will violate the assumption made in Theorem 1, and DECI cannot recover the ground-truth graphs and associated functional relationships. But the Bayesian setup allows us to leverage the help from the priors. From Figure 12 in appendix E.6, we can see that DECI cannot perform accurate ATE estimation and graph discovery due to unidentifiability. With PC prior (i.e. use PC discovered graph as prior with different strength), we can slightly improve the ATE and graph discovery but not much since PC also cannot identify the graph. But if the some true graph information is provided (i.e. informed prior), we can improve the performance of DECI.
> >
> > > I disagree with the statement. In conditional-independence-based methods, the learning guarantees ....
> >
> > - Thanks for the reviewer’s comments. Kernel-CI requires the functions to be in the RKHS space, but one cannot test whether this holds in practice, similar to our assumption 1 for Theorem 1. Although one can argue that RKHS is flexible enough to approximate continuous functions arbitrarily with characteristic kernels, DECI can also leverage the universal approximation property of neural networks.
> >
> > - We admit that we do not provide convergence rate of DECI but we do claim that under the assumptions of Theorem 1, DECI does recover the ground truth data mechanism. No latent confounder assumption is very common in many causal discovery literature [Glymour et al, 2019], which we do not think it is over idealized. For example, PC and Guo et al., 2021 mentioned by the reviewer also assume no latent confounders.
> >
> > - Regarding the model specification, both “f is in Sobolev spaces” and “DECI is correctly specified” are hard to verify in practice. To that end, they are equivalent in terms of practical meaning. Well-specification assumption is more general than “f is in Sobolev spaces”, and can be found in well-known literature in variational inference like [Wang et al, 2018; Kleijn et al, 2012; Van der Vaart, 2000]. In these literature, their definition about model well-specification is identical to ours.
> >
> > - Although there are also several works in variational inference to relax well-specification constraints and provide convergence rate [Kleijn et al, 2012], they cannot be directly applied to our setting due to the discrete nature of graphs. We want to emphasize that the theory is only part of the contribution of DECI, and so we think these additional theoretical guarantees are more suitable for a future work as they are independent interests to both causal discovery and variational inference communities.
> >
> >
> > Reference
> >
> > Glymour, C., Zhang, K. and Spirtes, P., 2019. Review of causal discovery methods based on graphical models. Frontiers in genetics, 10, p.524.
> >
> > Wang, Y. and Blei, D.M., 2018. Frequentist consistency of variational Bayes. Journal of the American Statistical Association.
> >
> > Kleijn, B.J. and Van der Vaart, A.W., 2012. The Bernstein-von-Mises theorem under misspecification.
> >
> > Van der Vaart, A.W., 2000. Asymptotic statistics (Vol. 3). Cambridge university press.

---

### Review · Reviewer_RTJh · 2024-03-29

**Summary Of Contributions:**

The paper proposes a unified framework that performs both causal discovery and causal inference simultaneously from observational data, namely deep end-to-end causal inference (DECI). This work  contributes to the field of causal inference by addressing a longstanding challenge that combines the causal discovery and causal inference processes within the same framework by the Bayesian graph structure. Key contributions of the paper include:

1. Introduction of an end-to-end non-linear additive noise model that utilizes neural network functional relationships for both causal discovery and inference, including the estimation of average treatment effects (ATE) and conditional ATE，.
2. Theoretical guarantees for the framework are provided, demonstrating that DECI can asymptotically recover the true causal graph and treatment effects, assuming the model is correctly specified with the true parameters.
3. DECI's performance is empirically validated through extensive experiments, showing its competitive or superior performance compared to existing methods in both causal discovery and casual effect estimation across a variety of synthetic, real and benchmark datasets.

**Audience:**

Yes

**Broader Impact Concerns:**

I don't have any concerns about the broader impact of this paper.

**Claims And Evidence:**

Yes

**Requested Changes:**

Please see the weakness above.

**Strengths And Weaknesses:**

## Strength:
1. I like the proposed framework which works for both the causal discovery and causal inference, where the causal discovery is via learning the variational distribution $q_{\phi}(G)$ by ENCO parameterisation, and the causal inference is estimated by constructing MLPs for each node with function $f_i(\mathbf{x})$ for characterizing $p_{\theta}(\mathbf{x}^n|G)$. This framework is a unified view of functional causal discovery including many previous literatures such as Notears, GAE and Grandag.
2. Further, the method is supported by theoretical analysis ad could be extended to more real scenarios such as mixed-type data and missing data.
3. This paper is excellently written, with a clear structure and methodology. The explanations across various aspects are commendable, and the overall framework is convincing. It is well-supported by overwhelming experiments and theoretical evidence, making for a very enjoyable reading experience.

## Weakness:
Nevertheless, while I really like the framework the paper proposed, I do have a few questions as list below.
### Major Comments:
1. Regarding the interpretation of the parameter $\theta$. There are two difference concepts here, one is the true parameter $\theta^*$ for the true function $f_i$ (if we assume it is parametric), and another is the estimated $\tilde{\theta}$ that is characterised by $\ell_i$ and $\zeta_i$. It seems the paper treats them as the same concept throughout the paper where the flexible parameterisation is matching with the true parameterisation. Also the same confusion rises up on the correct specification where $p_{\theta^*}$ would have the same structure as the MLPs and through maximising the ELBO. Then the MLP parameters will be converging to the true parameters with a large number of data. However, in most real cases it may not be true, do the authors have more comments on misspecified models where the functional estimation varies from the true estimates?
2. While I really appreciate the great efforts that author made on the performance evaluation, it would be interesting to see the discussions on the training sample sizes as they are fixed across some of the experiments. It would be interesting to add a few discussions on the effectiveness of the framework with different training sample sizes.
3. The framework consists of many surrogate components as listed below. It would be helpful to find out which components would lead to the most effective estimation for getting more insights. Also it would be good to know the training and inference complexity for different tasks.
    - Prior over $G$
    - Parameterization of $f_i$
    - Parameterization of $q_{\phi}$
    - $g_G$ with different basis-function linear models
    - Hyper-parameters of training of MLPs



### Minor Comments:
1. It seems that $f_i$ is assumed to be parametric at the first place. Any comments on the scenarios where $f_i$ is non-parameteric?
2. “A naïve implementation would require training 2D MLPs. Instead, we construct these MLPs so that their weights are shared across nodes…” This sentence is a bit confusing, do you mean instead of training 2D MLPs, we only train 2 MLPs for $\ell$ and $\zeta$, respectively across different nodes?
3. Abuse of notation where $g_G$ on (13) is the same as (7).
4. Some notations could be more clearly specified such as Hadamard product on (5), Fronbiues Norm on (6), and dirichlet function $\delta$.
5. It would be great to give more intuitions on the parameterization of ENCO in the context for $q_\phi$.

---

> ### Author Response · Authors · 2024-04-11
> **Authors' response 1**
>
> Thanks for the reviewer's valuable feedback, we will try to address the concerns in the following.
>
> 1. > Regarding the interpretation of the parameter $\theta$. There are two different concepts here ...
>
> In the current presentation, we do not explicitly differentiate the parameterization of the proposed DECI and ground-truth functional structures. This is because we assume the Assumption 1 (highlighted in the revised version) is satisfied to make sure Theorem 1 holds. We want to make one clarification: Assumption 1 only assumes the correct specification in the sample level, rather than the structural level. Namely, the ground truth functions can have different structures than DECI $l_i$ and $\zeta_i$. As long as DECI is expressive enough to represent the true function, Assumption 1 will hold. Another point worth mentioning is that we do not claim the convergence of the estimated $\theta$ (from maximizing ELBO) to the ground-truth $\theta*$. The parameter identification is in fact impossible because of the over-parameterized neural network. Theorem 1 only claims the identification of the distribution and graph. Therefore, it is possible that the estimated parameter $\theta$ is completely different from true $\theta*$, but results in the same observational distribution.
>
> When the model is misspecified (i.e. the true generation mechanism cannot be represented by DECI parameterization), Theorem 1 no longer holds. This means that there is no guarantee on the recovery of the true generation mechanism and graphs. There are some existing works on variational inference under the misspecified model [1, 2]. For example, [1] claims that even with model misspecification, the posterior will collapse to a point that minimizes the KL divergence between the resulting and ground-truth generation mechanism. However, their setup cannot be directly applied to graph distribution of DECI since they assume continuous random variables. We believe a similar guarantee should be possible for DECI as well, and we will leave that for future work.
>
> 2. > While I really appreciate the great efforts that author made on the performance evaluation, it would be interesting to see the discussions on the training sample sizes ...
>
> Thanks for pointing out this ablation study. In fact, Figure 6 in Appendix A.5 shows one important effect of changing training sample size. According to Theorem 1, when the training sample size increases, the uncertainty of the graph should decrease accordingly. Figure 6 confirms this by plotting the entropy of the variational graph distribution, which decreases with increasing sample size. So with small training sample size, the variational distribution will sample multiple diverse graphs, resulting in possible high variance inference performances.
>
> We have highlighted the relevant part in the revised paper.
>
> Reference:
>
> [1] Wang, Y. and Blei, D., 2019. Variational Bayes under model misspecification. Advances in Neural Information Processing Systems, 32.
>
> [2] Knoblauch, J., 2019. Frequentist consistency of generalized variational inference. arXiv preprint arXiv:1912.04946.
>
> [3] Annadani, Y., Rothfuss, J., Lacoste, A., Scherrer, N., Goyal, A., Bengio, Y. and Bauer, S., 2021. Variational causal networks: Approximate bayesian inference over causal structures. arXiv preprint arXiv:2106.07635.

---

> ### Author Response · Authors · 2024-04-11
> **Authors' response 2**
>
> 3. > The framework consists of many surrogate components as listed below...
>
> - Prior over $G$: It should not have a significant effect on the performance of DECI unless expert knowledge is incorporated (Appendix E.6). The augmented Lagrangian procedure on $\rho$ and $\alpha$ should have larger impact compared to the design of prior itself, since this controls the DAGness of the inferred graph distribution.
>
> - Parameterization of $f_i$: This will have an impact based on the capacity of $f$. For example, whether $f$ is linear or not will affect the model specification assumption (Assumption 1). When model is linear but the ground truth is nonlinear, worse performance is expected.
> - Parameterization of $q$: In general, the parameterization of the approximate graph posterior will also have an impact to the performance as well. In our work, we restrict the parameterization to be a Bernoulli distribution over edges, which does not model the edge co-dependencies. One can propose more complicated designs to incorporate more complex relationships, such as auto-regressive sampling [3]. Our approach can be easily adapted to such a design.
>
> - Different basis function $g_G$: The estimator capacity will also have an impact, since a good regression model can represent the conditional distribution with more accuracy, leading to an accurate CATE. However, this is based on the accuracy of graph discovery and function learning. Therefore, it should not have a larger impact compared to other design choices discussed above. Regarding the choice of the estimator, one can go beyond basis functions, e.g. neural network, but it requires additional training.
>
> - Hyper-parameters: In general, DECI is not very sensitive to hyper-parameter choices. However, the sparsity coefficient $\lambda_s$ in Eq.6 will impact the discovery performance. Setting it too large or too small will lead to sparse or dense graphs, respectively.
>
> - Noise distribution: In addition to these design choices mentioned by the reviewer, the noise distribution also has significant impact. Figure 4 shows that flexible noise distribution (i.e. Spline) tends to perform better in discovery compare to the simple Gaussian noise. For inference (Table 1), DECI with Spline noise also gives better accuracy compared to Gaussian noise.
>
> 4. > Minor comments: (1) non-parametric $f_i$; (2) Training 2D networks; (3) abuse of notations; (4) clear notations; (5) Intuition of ENCO parameterization ...
>
> - In this work, we always assume the function $f_i$ is parametric and is modelled by neural networks. The reason is that it naturally fits the proposed variational inference framework and allows efficient back-prop training. If the non-parametric function is used (e.g. Gaussian process), it is not trivial to (1) incorporate the graph, and (2) efficient training under large dataset.
>
> - Since we have $D$ nodes and each function $f_i$ consists of 2 MLPs ($l_i,\zeta_i$). So in total, we need to train $2D$ MLPs. To enable weight sharing, we incorporate an embedding $u_i$ associated to each node $i$ so that $f_i(\cdot) = f(\cdot,u_i)$. We have highlighted the relevant part in the revised version.
>
> - Thanks for spotting the symbol reuse. We have change to $h_G$ to indicate the conditional mean in Eq.13.
>
> - Thanks for your suggestion. We have add the symbol definition in the revised paper.
>
> - We have added relevant discussion about ENCO parameterization in the revised paper. In a nutshell, ENCO separates the edge existence and orientation probability, and set $v_{ij}=-v_{ji}$ for the orientation logits. Therefore, the two orientation probabilities ($\sigma(v_{ij})+\sigma(v_{ji})$) will always sum to one. A high probability direction indicates a low probability of the other direction. This can help to prevent the length-1 cycles. In addition, the existence probability is easier to learn compared to the orientation since it does not need to consider the dagness. Thus, this separation allows for better control of gradient.

---

### Author Response · Authors · 2024-04-11
**Authors' response**

We greatly appreciate the valuable feedbacks from the reviewers. We have revised the paper based on some suggestions. We use three colours (blue, red and green) to indicate the revisions, along with a margin note to indicate (1) which reviewer makes this suggestion; (2) type of revision; and (3) short summary of the revision.

- Blue means **revision**, where we revise the previous wording or make further clarification.
- Green means **highlight**, where we highlight the text to answer some of reviewers' concerns.
- Red means **addition**, where we add additional discussion or clarification based on the reviewers' feedbacks.

---

### Decision · Action_Editor_KhQh · 2024-05-01

**Recommendation:** Accept as is

**Comment:**

A majority of the reviewers supported accepting the paper but pointed out small revisions that could improve it. In particular, for the camera-ready version, reviewers asked that the following be included:

* A comment on whether causal effects are always identifiable in the learned graphs, and if not, what the algorithm does in the case where this fails.
* A comment on the sample efficiency of the method in relation to the literature, such as Jung et al., (2021b).
* An elaboration (in the main paper) on the difference between $\theta$ and $\theta^*$, potential model misspecification, and misestimation, both in the case where $\theta^*$ is unique and where it is not. Currently, there is only a brief comment in the appendix.

**Audience:**

Yes. The topic of the paper is of great interest and the contributions are substantial enough to be suitable for the TMLR audience.

**Claims And Evidence:**

Yes. Reviewers appreciated the substantial experimental validation of the methodology, as well as its theoretical justification.